EMBO
Molecular Medicine

# Comprehensive molecular characterization of collecting duct carcinoma for therapeutic vulnerability

Peiyong Guan [1,19], Jianfeng Chen [2,19], Chengqiang Mo[3,19], Tomoya Fukawa[4,19], Chao Zhang[5], Xiuyu Cai[2], Mei Li[2,6], Jing Han Hong[7], Jason Yongsheng Chan [8], Cedric Chuan Young Ng [9], Jing Yi Lee[9], Suet Far Wong[9], Wei Liu[9], Xian Zeng[2], Peili Wang[2], Rong Xiao [2], Vikneswari Rajasegaran[9], Swe Swe Myint[9], Abner Ming Sun Lim[9], Joe Poh Sheng Yeong[10,11], Puay Hoon Tan[11,12,13], Choon Kiat Ong [14], Tao Xu[15], Yiqing Du[15], Fan Bai [16], Xin Yao [5✉], Bin Tean Teh [1,7,9,17✉] & Jing Tan [2,9,18✉]

## Abstract

**Collecting duct carcinoma (CDC) is an aggressive rare subtype of kidney cancer with unmet clinical needs. Little is known about its underlying molecular alterations and etiology, primarily due to its rarity, and lack of preclinical models. This study aims to comprehensively characterize molecular alterations in CDC and identify its therapeutic vulnerabilities. Through whole-exome and transcriptome sequencing, we identified KRAS hotspot mutations (G12A/D/V) in 3/13 (23%) of the patients, in addition to known TP53, NF2 mutations. 3/13 (23%) patients carried a mutational signature (SBS22) caused by aristolochic acid (AA) exposures, known to be more prevalent in Asia, highlighting a geologically specific disease etiology. We further discovered that cell cycle-related pathways were the most predominantly dysregulated pathways. Our drug screening with our newly established CDC preclinical models identified a CDK9 inhibitor LDC000067 that specifically inhibited CDC tumor growth and prolonged survival. Our study not only improved our understanding of oncogenic molecular alterations of Asian CDC, but also identified cell-cycle machinery as a therapeutic vulnerability, laying the foundation for clinical trials to treat patients with such aggressive cancer.**

**Keywords** Collecting Duct Carcinoma; Whole Exome Sequencing; Transcriptome Profiling; Drug Screening; Cell-Cycle Machinery

**Subject Categories** Cancer; Urogenital System

## Introduction

Collecting duct carcinoma (CDC) is one of the most aggressive kidney cancer, which accounts for about 1% of all kidney cancers (Suarez et al, 2022). NCCN (National Comprehensive Cancer Network) Guidelines® for kidney cancer (version 1.2024) recommends surveillance or clinical trial, after CDC patients' radical/partial nephrectomy. Even though early surgical treatment can improve patient survival, CDC patients usually present with higher grade, and advanced stage, where as high as 70% of the patients present with metastases, having only median overall survival of 13 to 17 months, and 5-year cancer-specific survival of only 30% (Sui et al, 2017; Tang et al, 2021). There is no effective targeted therapy for CDC (Panunzio et al, 2023; Xie et al, 2022). Chemotherapy and immunotherapy were reported to achieve complete or partial responses; however, these treatments were primarily in case reports or small cohorts with limited benefits, highlighting the necessity of understanding the molecular basis of CDC (Dason et al, 2013; Funajima et al, 2023; Guillaume et al, 2022; Oudard et al, 2007; Procopio et al,

¹Genome Institute of Singapore (GIS), Agency for Science, Technology and Research (A*STAR), 60 Biopolis Street, Genome, Singapore 138672, Republic of Singapore. ²State Key Laboratory of Oncology in South China, Guangdong Provincial Clinical Research Center for Cancer, Collaborative Innovation Center for Cancer Medicine, Sun Yat-sen University Cancer Center, Guangzhou, P. R. China. ³Department of Urology, The First Affiliated Hospital of Sun Yat-sen University, Guangzhou, PR China. ⁴Department of Urology, Tokushima University Graduate School of Biomedical Sciences, Tokushima, Japan. ⁵Department of Genitourinary Oncology, Tianjin Medical University Cancer Institute and Hospital, National Clinical Research Center of Cancer, Tianjin Key Laboratory of Cancer Prevention and Therapy, Tianjin's Clinical Research Center for Cancer, Tianjin, P. R. China. ⁶Department of Pathology, Sun Yat-sen University Cancer Center, Guangzhou, PR China. ⁷Cancer and Stem Cell Biology Programme, Duke-NUS Medical School, Singapore, Republic of Singapore. ⁸Division of Medical Oncology, National Cancer Centre Singapore, Singapore, Republic of Singapore. ⁹Laboratory of Cancer Epigenome, Division of Medical Sciences, National Cancer Centre Singapore, Singapore, Republic of Singapore. ¹⁰Institute of Molecular and Cell Biology (IMCB), Agency of Science, Technology and Research (A*STAR), Singapore, Republic of Singapore. ¹¹Department of Anatomical Pathology, Singapore General Hospital, Singapore, Republic of Singapore. ¹²Division of Pathology, Singapore General Hospital, Singapore, Republic of Singapore. ¹³Luma Medical Centre, Singapore, Republic of Singapore. ¹⁴Lymphoma Genomic Translational Research Laboratory, National Cancer Centre Singapore, Singapore, Republic of Singapore. ¹⁵Department of Urology, Peking University People's Hospital, Beijing 100044, China. ¹⁶Biomedical Pioneering Innovation Center (BIOPIC), Beijing Advanced Innovation Center for Genomics (ICG), School of Life Sciences, Peking University, Beijing 100871, China. ¹⁷SingHealth/Duke-NUS Institute of Precision Medicine, National Heart Centre Singapore, Singapore, Republic of Singapore. ¹⁸Hainan Academy of Medical Science, Hainan Medical University, Haikou, PR China. ¹⁹These authors contributed equally: Peiyong Guan, Jianfeng Chen, Chengqiang Mo, Tomoya Fukawa. ✉E-mail: yaoxin@tjmuch.com; teh.bin.tean@singhealth.com.sg; tanjing@sysucc.org.cn

2022; Thibault et al, 2023; Yasuoka et al, 2018). Such unmet clinical need requires more comprehensive molecular characterizations of CDC.

Molecular alterations in CDC have been investigated through genome and transcriptome studies. Genomic characterizations of CDC are still ongoing with limited consensus. Pal et al, studied mutations in 17 CDC FFPE samples using targeted panels of cancer-related genes and found recurrent mutations in NF2 (29%), SETD2 (24%), SMARCB1 (18%), and CDKN2A (12%) (Pal et al, 2016). Wang et al, studied 7 CDCs but found no recurrent single nucleotide variant, except for MLL, but homozygous deletions of CDKN2A in three samples (Wang et al, 2016). Metastatic CDC harbored more mutations in SMARCB1, NF2, RB1, and RET, compared to clear cell renal cell carcinoma (ccRCC) (Bratslavsky et al, 2021). Transcriptome data for the CDC are similarly limited to a few studies on Caucasian patients. Malouf et al, profiled 11 CDC patients, identifying distal convoluted tubules as its cells of origins (Malouf et al, 2016). Wach et al, studied 2 CDCs, and found overexpression of KRT17, which correlated with poorer survival in ccRCC (Wach et al, 2019). Gargiuli et al, profiled six CDC samples and discovered two CDC subtypes with different cell signaling, metabolic and immune profiles (Gargiuli et al, 2021). Wang et al, found upregulation of SLC7A11, a cisplatin resistance-associated gene, in 80% (four out of five) of the CDC samples checked (Wang et al, 2016). Msaouel et al, profiled nine CDCs, but the study mainly focused on renal medullary carcinoma (Msaouel et al, 2020a). Frequent overexpression of HER2 protein was observed (23%, 6/26) (Costantini et al, 2020). Together, our understanding of transcriptomic dysregulation in the CDC remains limited, in part due to the small sample sizes and heterogeneity of the CDC.

Therefore, existing CDC studies have their limitations: (1) most data on CDC are for Caucasian patients, while CDC in Asians has not been molecularly characterized; (2) besides sample size, CDC therapeutics design is also hampered by the scarcity of preclinical models, including cell lines and patient-derived xenograft (PDX) models for drug screening and testing (Wu et al, 2009).

To address these limitations, in this study, we analyzed the genome and transcriptome in our new Asian CDC cohort ($n = 14$, 13 normal-matched tumors and 1 PDX model). Besides a new CDC cell line, to our knowledge, we are the first to establish a CDC PDX model for drug screening and testing, leading to the identification and successful validation of a CDK9 inhibitor (LDC000067) that specifically inhibited CDC growth, providing a potential therapeutic strategy for CDC treatment.

# Results

Our Asian CDC cohort consisted of 14 patients, eight males and six females, with a mean age of 60. The majority (11/14) of the patients are Chinese, two were Japanese, and one with no ethnic information available. In total, 13 fresh-frozen normal-matched CDC tumors were analyzed by WES, eight of which were also profiled with RNA-seq. One PDX (CDC1) and its cell line models were established for drug screening and validations. (Appendix Tables S1, S2).

## Recurrent somatic mutations in Asian CDC and mutational signatures indicating possible etiology

We first cataloged the mutational landscape of Asian CDC (Fig. 1). The average tumor mutational burden (TMB) of the normal-matched tumors is 1.86 per MB, ranging from 0.12 to 7.02 per MB

(Fig. 1A; Appendix Fig. S1A). The most commonly mutated oncogenes (Fig. 1B; Appendix Fig. S1B; Dataset EV1) include TP53 mutations (Appendix Fig. S1C,D) in 23% (3/13) of the patients; NF2 mutations in 15% (2/13) of the patients; and KRAS G12A/D/V hotspot mutations in 25% (3/13) of the patients (Appendix Fig. S1E,F). Our cohort also has one patient with FH heterozygous deletion (Appendix Fig. S1G). When somatic mutations are grouped by the oncogenic pathways they are involved in, the Hippo signaling pathway was mutated in 46% of the patients, followed by the RTK-RAS pathway (46%). Other mutated pathways include NOTCH (31%), TP53 (31%), MYC (15%), PI3K (15%), TGF-Beta (15%), and WNT (8%) (Fig. 1B).

No clear association between patient demographics and mutation type was observed (Fig. 1C,D). The types of single base substitutions differ among patients, with T04, T05, and T10 mostly showing T > A mutations (Fig. 1D,E; Appendix Fig. S1H). Mutational signature analysis revealed these three patients carried mutational signature SBS22, caused by exposure to aristolochic acid (AA), a known Group 1 carcinogen (Fig. 1E). AA exposure is known to cause higher mutational load and more neoantigens, which was confirmed by our neoantigen analysis, where the number of strong binder neoantigens generally correlated with the mutational load (Fig. 1F). Other mutational signatures are associated with aging (SBS1 and SBS5), activity of AID/APOBEC family of cytidine deaminases (SBS2 and SBS13), and occupational exposure to haloalkanes (SBS42). No focal amplification was identified whereas focal deletion was found at 9p21.3 ($q$ value = 0.071), where CDKN2A and CDKN2B genes are located, though with marginal statistical significance. (Appendix Fig. S1I–K). Read coverage at the CDKN2A and CDKN2B loci showed that tumor samples had relatively lower coverage, as compared to the normal samples, supporting shallow deletions (Appendix Fig. S1L). However, focal deletion in the 9p21.3 did not reduce expression levels of CDKN2A or CDKN2B, as they were increased by 3.37 and 1.39 log2-folds in the tumors respectively (Appendix Fig. S1M,N).

## Genomic mutation and associated transcriptomic alteration inform precision therapy

Leveraging our matched WES and RNA-seq data, we next investigated whether the frequently mutated Hippo and RTK-RAS pathways, as well as AA mutations, lead to targetable transcriptomic alterations (Fig. 2A). To mitigate our relatively small sample size, we excluded outlier samples that did not cluster transcriptomically with the majority of the samples sharing the same mutated pathway. This allowed us to study relatively more homogeneous groups of samples (Appendix Fig. S2).

We first observed that mutations in the Hippo pathway led to downregulation of the cell cycle and MYC-target, compared to Hippo wild-type samples. Since the cell cycle pathway was upregulated in CDC, independent of Hippo mutations, this implied different underlying mechanisms drive cell cycle progression. Different from the Hippo pathway-mutant tumors, the wild-type tumors showed induced immune-related pathways, such as interferon alpha/gamma response (Fig. 2B,C). As expected, KRAS hotspot mutations led to dysregulation of known KRAS-targeted genes (Fig. 2D,E). Interestingly, patients harboring KRAS mutations had worse survival (Fig. 2F, Cox model, $p = 0.0039$). Patients

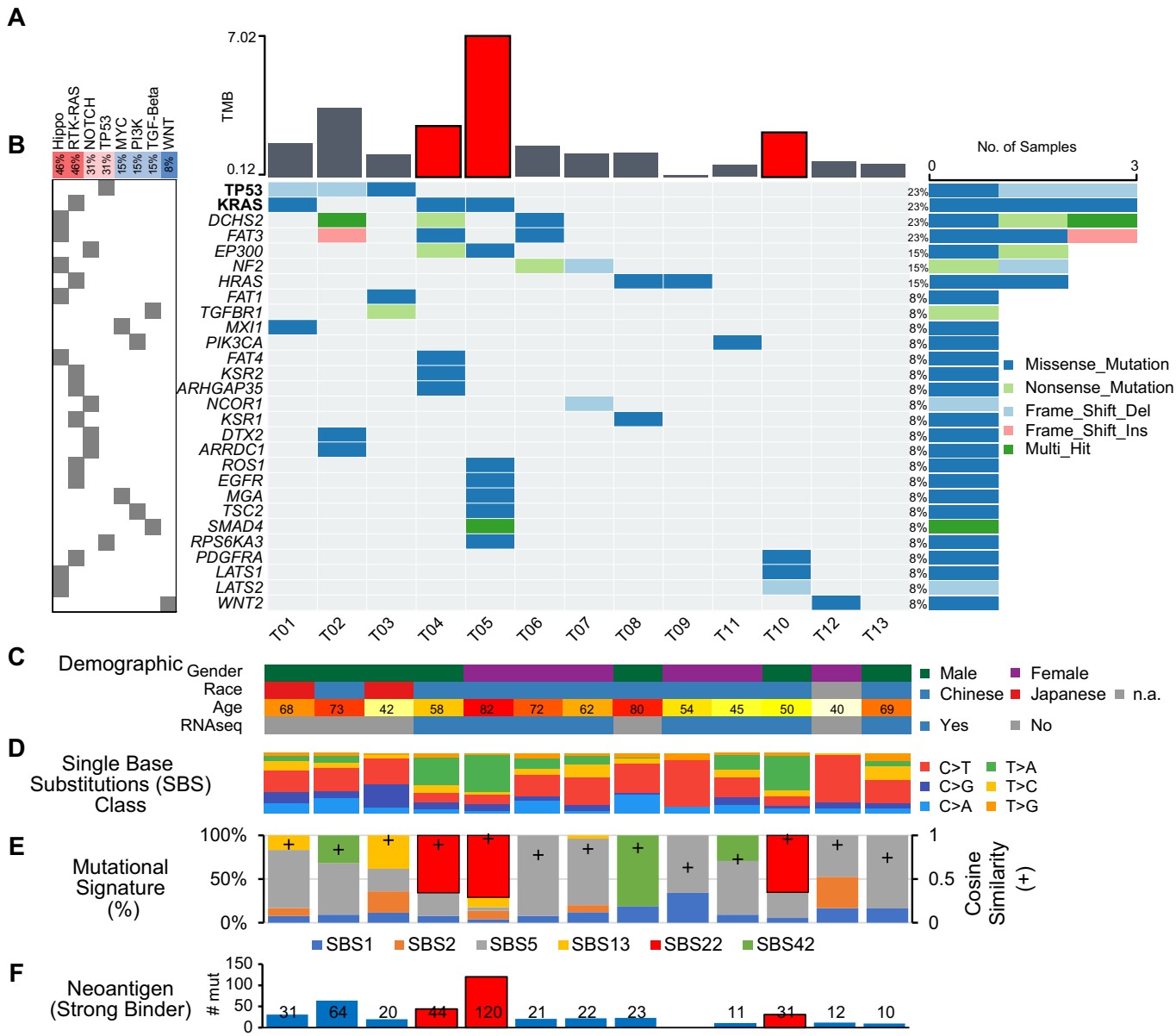

**Figure 1. Genomic landscape of collecting duct carcinoma.**

(A) Tumor mutation burden (TMB). (B) Mutations in oncogenic pathways and percentage of samples in which the pathways are mutated in (left). (C) Demographic information of the patients. (D) Single base substitutions (SBS) Class. (E) Mutational signature (%) in each patient. Cosine similarity: accuracy metric between 0 and 1 for the reconstruction of the original mutational catalog. (F) Neoantigen (strong binder) prediction. Source data are available online for this figure.

with AA-signatures exhibited an upregulated NFκB pathway, consistent with a higher number of CD8+ naive T−cells and Th1 cells (Fig. 2G–I). Because of the co-occurrence of RTK-RAS mutation and AA signature, AA-positive tumors also showed altered expressions in KRAS-targeted genes (Figs. 1B, 2A,H). However, mutations in KRAS are C > G/A/T, different from the typical T > A mutations caused by AA (Poon et al, 2013), indicating the involvement of multiple mutational processes.

Besides pathways related to specific mutations, epithelial and mesenchymal transition (related to cancer invasion) and cell cycle-related pathways were commonly upregulated in CDC (Fig. 2C,D,H).

Therefore, we hypothesized that, although CDC tumors exhibit heterogeneity in terms of somatic mutations and associated transcriptomic dysregulation, cell cycle-related pathways may be a commonly induced pathway in CDC that is potentially targetable. We next studied cell-cycle pathways in CDC in depth.

## Cell cycle pathway is commonly upregulated in both Asian and Caucasian CDC

At the whole transcriptome level, Principal Component Analysis (PCA) showed that CDC tumors ($n = 8$) were clearly separated

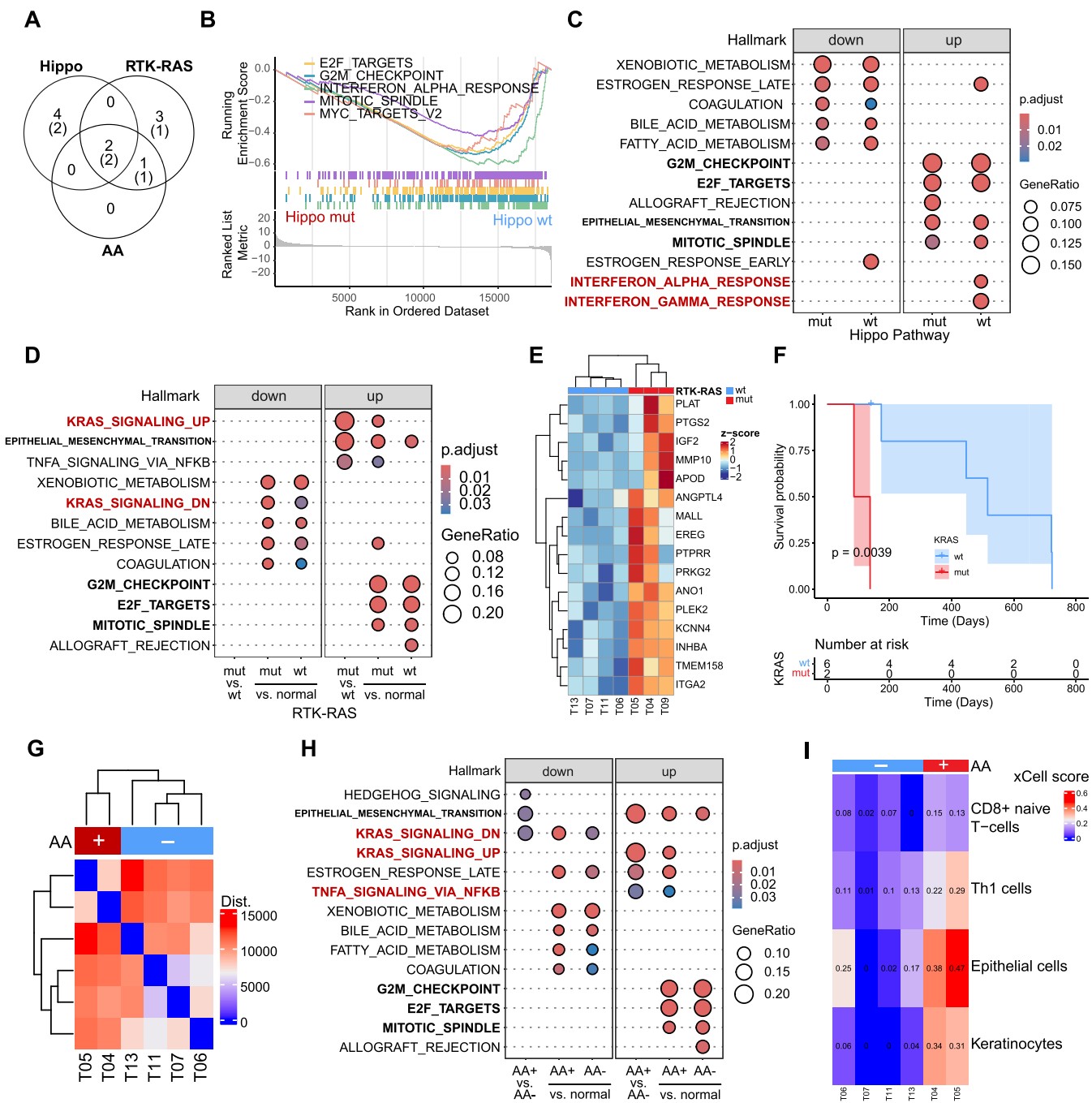

**Figure 2. Genomic mutation and associated transcriptomic alteration inform precision therapy.**

(A) Overlapping of patients with different oncogenic mutations. Numbers in brackets indicate the number of cases with matched RNA-seq data. (B) Gene set enrichment analysis (GSEA), comparing Hippo pathway mutant (mut) and wild-type (wt) samples. (C) Enrichment of differentially expressed genes between Hippo mut, Hippo wt tumor, and normal samples. Hypergeometric test with Benjamini–Hochberg (BH) adjustment, where reference background is the total number of genes in the Hallmark collection. (D) Enrichment of differentially expressed genes between RTK-RAS pathway mutant (mut) and wild-type (wt) samples, as well as normal samples. Hypergeometric test with Benjamini–Hochberg (BH) adjustment, where reference background is the total number of genes in the Hallmark collection. (E) Heatmap showing expression levels of genes upregulated in HALLMARK_KRAS_SIGNALING_UP gene set, comparing RTK-RAS mutant (mut) with wild-type (wt) tumors. (F) Kaplan–Meier plot based on KRAS genotype (wt: patients with wild-type KRAS, $n = 6$; mut, patients with KRAS mutation, $n = 2$). P value: log-rank test. (G) Whole transcriptome of AA-positive (+) and AA-negative (−) samples. Dist.: Poisson dissimilarity matrix. (H) Enrichment of differentially expressed genes between AA-positive (+), AA-negative (−) and normal samples. Hypergeometric test with Benjamini–Hochberg (BH) adjustment, where reference background is the total number of genes in the Hallmark collection. (I) Cell type enrichment analysis. Only those showed significant differences between AA-positive ($n = 2$) and AA-negative ($n = 3$) samples are shown (one tail $t$-test, $p$ value ≤0.05). $p = 1.37E-02$, $1.50E-02$, $1.17E-04$, and $1.04E-02$ for CD8+ naive T-cells, Epithelial cells, Keratinocytes, and Th1 cells respectively. Source data are available online for this figure.

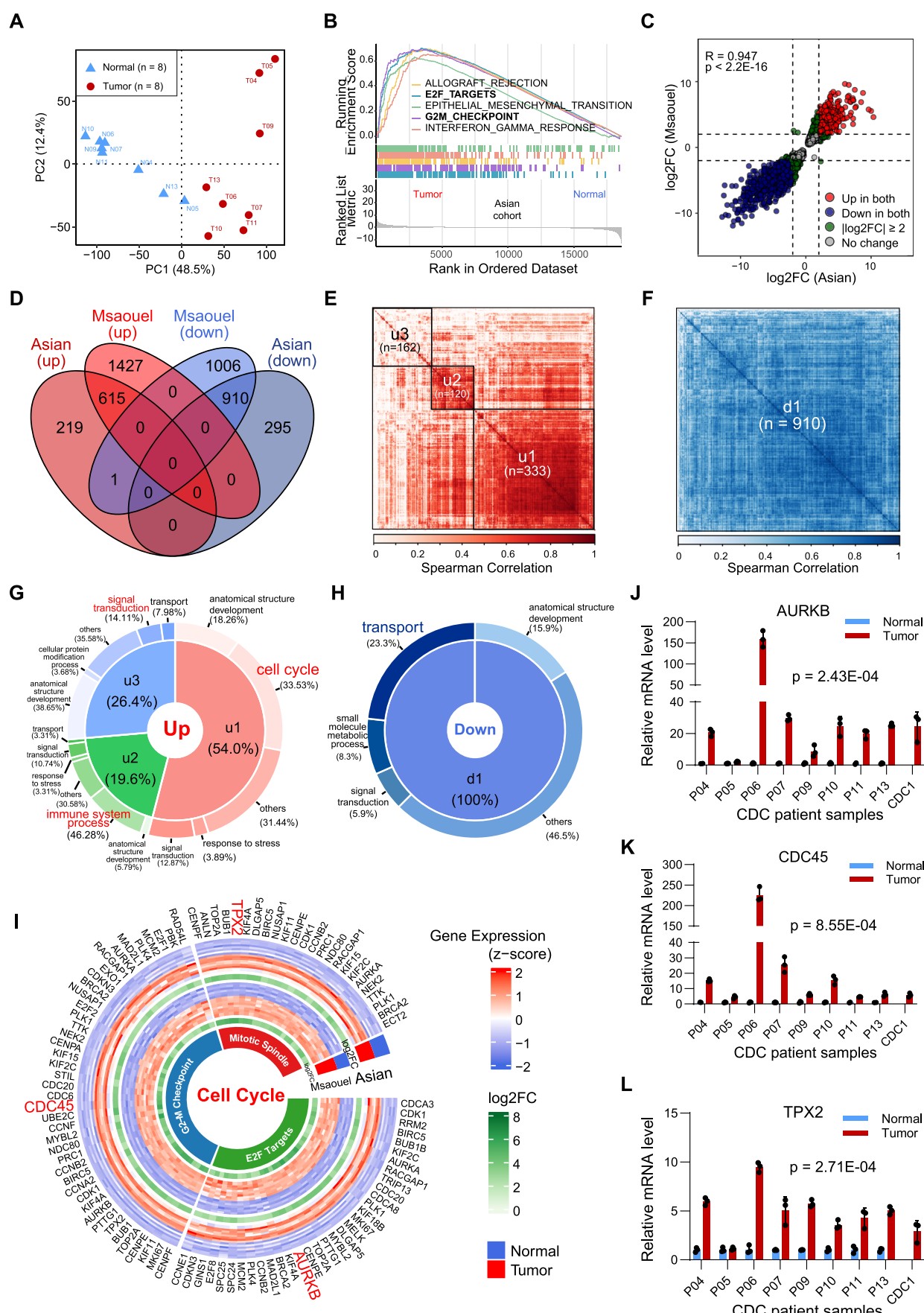

**Figure 3.  Identifying drivers for CDC tumorigenesis by transcriptomic profiling.**

(A) Principal component analysis (PCA) of Asian cohort (*n* = 8 pairs). (B) Gene set enrichment analysis (GSEA), top five enriched Hallmark pathways shown.
(C) Correlation of log2 fold changes (log2FC) in Asian cohort and Msaouel Caucasian cohort. R: Pearson correlation coefficient; *p* < 2.2E-16: correlation test *p* value.
(D) Overlapping of differentially expressed genes in Asian cohort and Msaouel Caucasian cohort. Cutoff values: *p* value ≤0.01 and |log2FC|≥ 2. *P* values for overlapping of
up-/down-regulated genes in Asian and Caucasian cohorts <2.2E-16, Fisher's Exact Test. (E) Correlation coefficient (Spearman, squared) of the z-score normalized gene
expression levels of commonly upregulated genes. (F) Correlation coefficient (Spearman, squared) of the z-score normalized gene expression levels of commonly
downregulated genes. (G) Gene ontology analysis for commonly upregulated gene clusters. (H) Gene ontology analysis for commonly downregulated genes.
(I) Subcategorizing commonly upregulated cell cycle genes into different processes. (J–L) Real-time qRT-PCR (real-time quantitative reverse transcription PCR) validation
of selected genes, AURKB, CDC45, and TPX2. For each sample, data were representative of three independent experiments. Data were presented as the mean ± SD. *P*
value: paired *t*-test of average log2 of relative mRNA level (*n* = 3). Source data are available online for this figure.

from the normal kidney samples by the first principal component (PC1). CDC1 (PDX) resembled the CDC tumors (Fig. 3A).

Next, we examined the altered pathways. Cell cycle-related pathways E2F_Targets (normalized enrichment score (NES) = 2.73, *q* value = 2.81E-10) and G2M_Checkpoint (NES = 2.69, *q* value = 2.81E-10) were upregulated in tumors (Fig. 3B; Appendix Fig. S3A–C). Immune responses-related pathways, such as interferon-gamma response (Fig. 3B, NES = 2.54, *q* value = 2.81E-10) and TNFα signaling via NFκB (Appendix Fig. S3C, NES = 2.54, *q* value = 2.11E-10), were also induced in CDC.

Fold changes of differentially expressed genes (DEGs) in our Asian cohort significantly correlated with those in the Msaouel Caucasian cohort (Msaouel et al, 2020a) (Fig. 3C, *R* = 0.947 and *p* value = 0.0). There were 835 and 2042 upregulated genes in the two cohorts respectively, of which 615 are common; and there are 1205 and 1917 downregulated genes in the two cohorts, with 910 shared ones (Fig. 3D).

Correlation analysis of commonly upregulated genes identified three tight genes clusters, namely u1 (*n* = 333), u2 (*n* = 120), and u3 (*n* = 162) (Fig. 3E). Similarly, downregulated genes formed a large cluster d1 (*n* = 910) (Fig. 3F). Gene ontology analysis found that 33.53% of u1 genes were cell cycle-related (Fig. 3G); and 46.28% of the u2 genes were involved in immune system process and 14.11% of the u3 genes were related to signal transduction. On the other hand, the commonly downregulated genes were enriched in transport (23.3%) (Fig. 3H). Further analyses of the cell cycle genes found they were associated with E2F targets, G2-M checkpoint, and mitotic spindle (Fig. 3I). Expression of selected genes, AURKB, CDC45, and TPX2 were validated in nine CDC patient tissues using real-time qRT-PCR (Fig. 3J–L). Therefore, through transcriptome analysis, combined with published data, we found that cell cycle pathways might serve as drivers for CDC tumorigenesis.

## Drug screening identified CDK9 inhibitor, LDC000067, specifically suppressed CDC

For drug screening and testing, we established one PDX and primary cell line from patient CDC1's spinal metastasis (51 yr, Chinese, male) (Fig. 4A). CDC1 carries missense mutations in the FAT1 and LATS1 genes of the Hippo pathway (Dataset EV1). To further confirm the identity of the PDX (CDC1) sample as CDC, we first conducted immunohistochemistry (IHC) staining for PAX8 and CK19 in both CDC1 patients and PDX samples. The results showed that strong positive expression of PAX8 and CK19 in both samples, indicating these samples are real CDC (Fig. 4B). We next confirmed that CDC1 cell line transcriptionally resembled CDC tumors, distinct from normal kidneys, though some of the tumors,

such as T10, showed relatively lower level of cell-cycle pathways, highlighting tumor heterogeneity (Fig. 4C). The primary cell line CDC1 was screened using the drug library containing 130 compounds (1 μM) targeting cell cycle regulators for 96 h. Five drugs (Indisulam, Rigosertib, LDC000067, THZ1, and Briciclib) had an inhibition rate of at least 70%, and the top ten compounds were mostly CDK inhibitors (Fig. 4D,E).

We then evaluated the drugs' nephrotoxicity and specificity for CDC, by testing them in one normal immortalized kidney cell line (HK-2) and two ccRCC cell lines (A-498 and 786-O) (Fig. 4F). LDC000067, a CDK9 inhibitor, was selected for further validation due to its specific inhibition of CDC1 cell line by 74.86% and low nephrotoxicity with inhibition rate of only 3.15% on HK-2. We further showed that LDC000067 achieved dosage-dependent inhibition of CDC1 with an IC50 of 1.46 μM (Fig. 4G); and treating CDC1 with LDC000067 at 1 μM concentration significantly inhibited the cell growth (Fig. 4H, *n* = 5, *p* value = 3.30E-05, two-way ANOVA).

Therefore, LDC000067 can specifically inhibit CDC growth, with little toxicity to the normal kidney cell line.

## CDK9 inhibitor LDC000067 improved the survival of the CDC PDX model

After confirming the tumor inhibitory effect of LDC000067 in vitro, we used our PDX model to study its efficacy in vivo.

Compared to the vehicle treatment, LDC000067 significantly suppressed the tumor growth in the mice (Fig. 5A, *p* value = 1.1E-07, two-way ANOVA), thereby extending the median survival of the mice by 21 days (53.5 vs. 32 days, *p* value = 3.00E-06, log-rank test, Fig. 5B). Importantly, consistent with in vitro observations, LDC000067 was well tolerated, as reflected by the relatively constant body weight of the mice (Fig. 5C). Ki67 staining showed that LDC000067-treated tumors had significantly fewer number of tumor cells, compared with the vehicle-treated controls (Fig. 5D,E, *n* = 5, *p* value = 6.80E-06, two-sided *t*-test, equal variance). Importantly, induced genes (AURKA and TPX2) in CDC were downregulated upon drug treatment (Fig. 5D,F,G). Reduced p-RB1 staining upon LDC000067 treatment also indicated suppressed cell cycle progression (Fig. 5D,H).

In summary, we successfully demonstrated the tumor suppressive effect of LDC000067 in the PDX model.

## Discussion

CDC is an aggressive and rare subtype of kidney cancer with no effective targeted therapy. To date, the etiology of the disease is

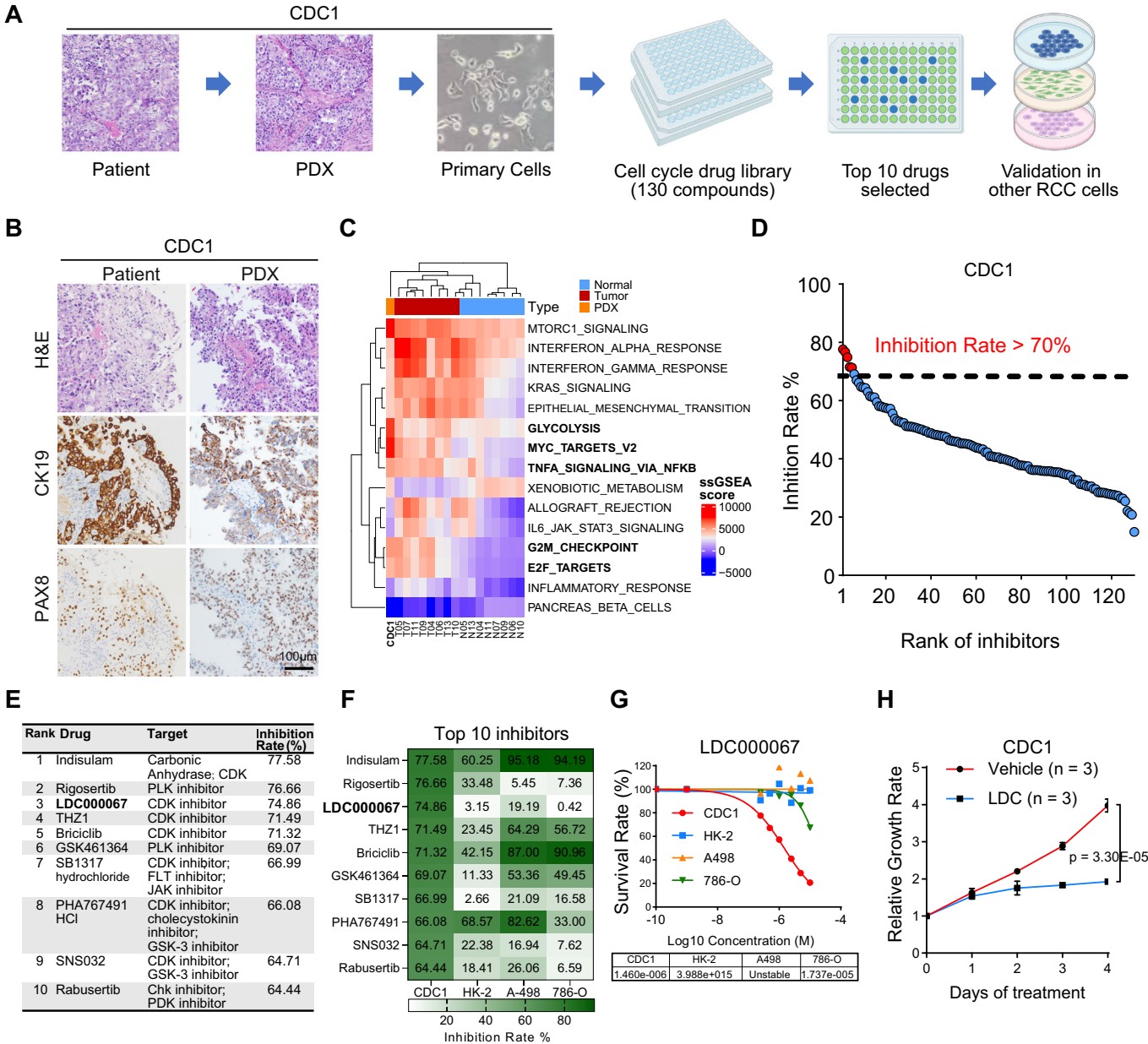

**Figure 4. CDK9 inhibitor LDC000067 specifically suppressed CDC rather than other RCC.**

(A) Workflow for establishing PDX and primary cell line; and screening for potential drug targets. (B) Hematoxylin and eosin (H&E), CK19, and PAX8 staining of CDC1 patient and PDX samples. (C) Single-sample GSEA (ssGSEA) score for all RNA-seq samples. The top 15 cell types (ranked by the variance of ssGSEA scores) are shown. CDC1 (PDX) clusters together with tumors, different from normal samples. (D) Overall ranking of the 130 small molecule compounds in the drug library by their inhibition rates. (E) Top ten drug candidates with the highest inhibition rates. (F) Testing of the top ten drug candidates at 1 μM concentration in clear cell RCC cell lines (A-498 and 786-O) and normal immortalized kidney cell line (HK-2). (G) Dosage-dependent response of CDC1 to CDK9 inhibitor LDC000067 and its IC50s for different cell lines. The cells were treated with LDC000067 for 96 h. (H) Relative growth rate of CDC1 when treated with LDC000067 at 1 μM concentration. Data were presented as the mean ± SD (n = 3). P value = 3.30E-05, two-way ANOVA. Source data are available online for this figure.

largely unknown, with patchy evidence for recurrent somatic mutations, primarily discovered in the Caucasian population. To date, only one other genomic study focused on Chinese CDC patients (*n* = 10) using formalin-fixed, paraffin-embedded (FFPE) samples, which found a similar TMB with ours (1.37 vs our 1.86 per MB) (Zhang et al, 2022a). Zhang et al, identified many focal amplifications and deletions, among which 9p21.3, where CDKN2A

and CDKN2B genes are located, is the only focal deletion found in our cohort. However, given the discordant gene expression changes in CDKN2A and CDKN2B, the role of 9p21.3 deletion in CDC tumorigenesis remains to be investigated. We hypothesize the increased CDKN2A and CDKN2B gene expressions were downstream dysregulation caused by mutations in oncogenic pathways, consistent with the induced cell-cycle pathway observed. Such

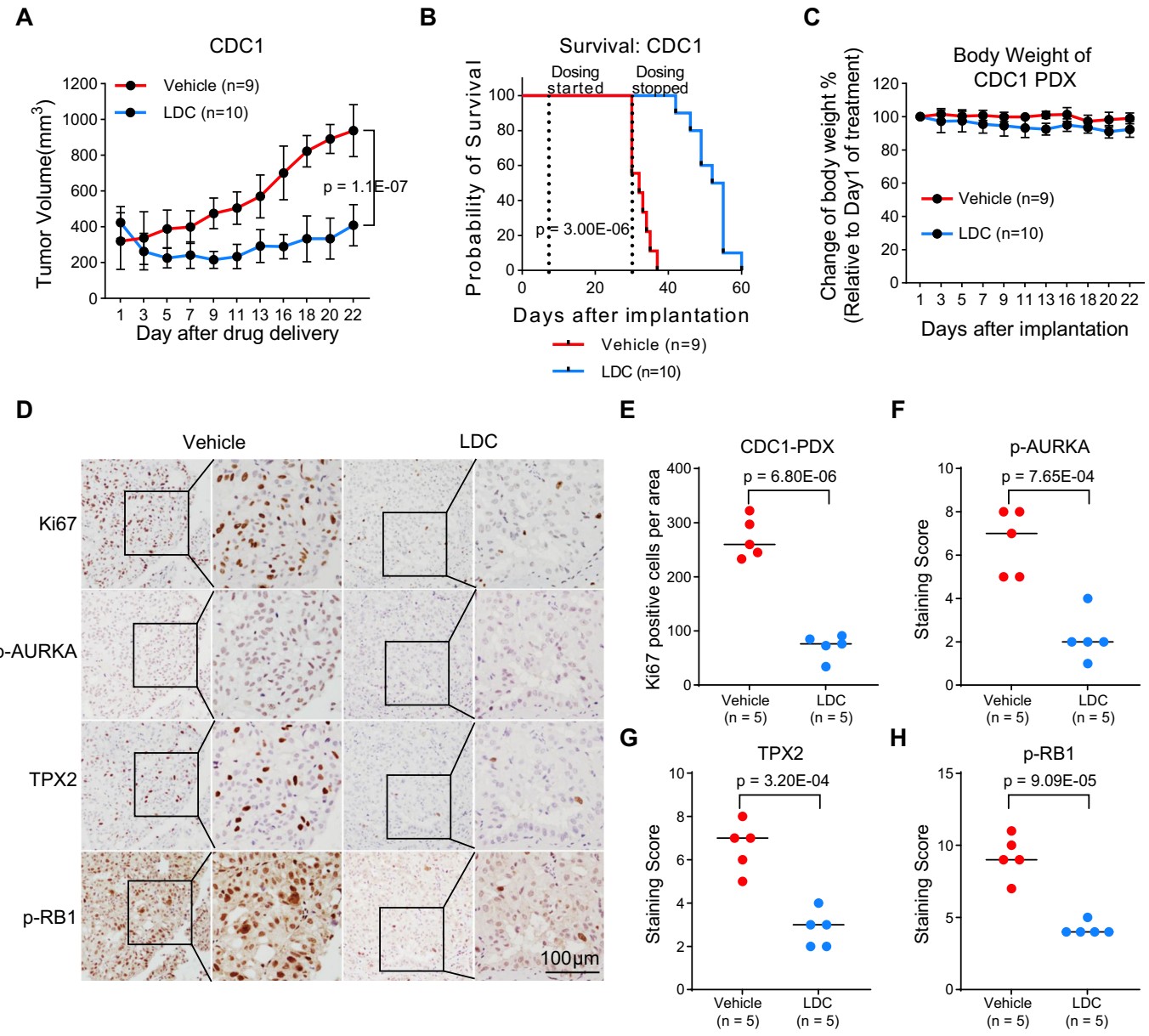

**Figure 5. CDK9 inhibitor LDC000067 prolonged the survival in the CDC PDX model.**

(A) Tumor volumes for CDC PDX models treated with vehicle control (*n* = 9 mice) or CDK9 inhibitor LDC000067 (LDC, *n* = 10 mice) (*p* value = 1.1E-07, two-way ANOVA). Data were presented as the mean ± SD. (B) Survival analysis for mice treated with LDC (*n* = 10 mice), compared with those treated with vehicle control (*n* = 9 mice) (*p* value = 3.00E-06, log-rank test). (C) Body weight of the CDC1 PDXs. Vehicle control (*n* = 9 mice) and LDC (*n* = 10 mice). Data were presented as the mean ± SD. (D) Ki67, p-AURKA, TPX2, and p-RB1 staining of selected PDX tumors treated with vehicle control or LDC. (E–H) Ki67 positive cells per area, a staining score of p-AURKA, TPX2, and p-RB1 respectively, comparing PDX tumors treated with vehicle control or LDC (*n* = 5 mice, two-sided *t*-test, equal variance). Source data are available online for this figure.

extensive focal changes in Zhang et al, cohort might be related to the nature of the FFPE samples. In terms of somatic small variants, TP53 was the only common top frequent mutations, but with different frequency (10% versus 23% in our cohort). The low concordance of the genomic changes indicates the heterogeneity of CDC tumorigenesis, which requires orthogonal characterizations, such as transcriptomic profiling.

To our knowledge, our study is the first integrative analysis of matched genome and transcriptome of Asian CDC. With the

matched genomic and transcriptomic data, we not only identified novel genomic mutations but also their downstream effects at the transcriptome level.

The genomic landscape of the Asian CDC we described may have important clinical implications. Recurrent TP53 and KRAS hotspot mutations in Asian CDC patients and their associated pathway dysregulations at the transcriptome levels could be exploited for precision therapy. TP53 mutated tumors can potentially be treated with various FDA-approved drugs (Hassin

and Oren, 2023). KRAS, a GTPase that regulates cell proliferation, frequently mutates in its codon 12 in multiple cancers, including our current Asian CDC cohort, where the patients were shown to have poor survival outcomes. Therefore, the recently developed KRAS hotspot mutation inhibitor, such as sotorasib (AMG 510) could potentially be used in treating these KRAS-mutant CDC patients (Canon et al, 2019; Kim et al, 2023; Zhang et al, 2022b). Because of our small sample size, and the wide confidence interval, especially for KRAS wild-type patients, how or whether the KRAS hotspot mutations lead to poorer survival requires further investigations.

Oncogenes in our Asian cohort showed different mutation frequencies than the Caucasian in Pal et al, (Pal et al, 2016), although not statistical significant due to sample size (one-sided Fisher's Exact Test). For example, mutation frequency of TP53 (23%, 3/13) is more than triple the 6% (1/17) reported by Pal et al, (*p* value = 0.20). NF2 mutated in 15% (2/13) of the patients, less than half the 29% (5/17) in Pal et al, (*p* value = 0.33). KRAS G12A/D/V hotspot mutations, not reported in Pal et al, occurred in 23% (3/13) of the patients (*p* value = 0.07). While such difference could be partly contributed by the different genomic backgrounds, environmental factors may play an equally important role, as suggested by the presence of mutational signature SBS42, caused by occupational exposure to haloalkanes. As we do not have occupational data on patients, the role of haloalkane exposure on CDC tumorigenesis requires further study, although it was previously found in printing workers who developed cholangiocarcinoma (Alexandrov et al, 2020).

Interestingly, we also found AA mutational signature (SBS22) in our Asian CDC patients, highlighting a geologically specific disease etiology like other Asian cancers (Das et al, 2022; Senkin et al, 2024). Because of the higher mutational burden and more prominent neoantigen presentations induced by AA exposure, this subgroup of CDC may respond to immune checkpoint blockade therapy, which is also supported by the relatively higher number of CD8+ naïve T−cells and Th1 cells observed in the AA+ tumors in our cohort.

By integrating Asian and Caucasian CDC transcriptome data, we identified that cell cycle, and immune response pathways to be commonly upregulated. Reassuringly, our data are consistent with previously published results (Gargiuli et al, 2021). These findings inform new therapeutic strategies for CDC, through cell cycle inhibition or utilization of immunotherapy; and the efficacy of immunotherapy has been reported in various case reports (Pyrgidis et al, 2023).

To facilitate drug screening and testing, we established CDC cell line and PDX models, as important resources for further drug development. To our knowledge, our PDX model is the first in the world, and the CDC cell line is one of the three cell lines available (Wu et al, 2009). Through drug library screening on our newly established CDC cell line, we found CDK9 inhibitor LDC000067 effectively inhibited the growth of CDC tumors, sparing normal kidney cells. This is the first in vivo validated potential CDC drug candidate.

Notably, epigenomic studies of CDC are still absent. This is mainly because the disease is rare and archived specimens are stored in FFPE (formalin-fixed paraffin-embedded) blocks, on which it is still technically challenging to profile genome-wide histone modifications.

To our knowledge, our study is the first multi-omics study of the Asian CDC. It not only improved our fundamental understanding of molecular alterations in CDC but also identified and validated cell cycle pathway inhibition as a viable therapeutic strategy. However, our CDC cell line model cannot possibly cover the entire heterogeneous spectrum of CDC. Further studies in more preclinical models are required before its clinical usage.

# Methods

**Reagents and tools table**

| Reagent/Resource | Reference or source | Identifier or catalog number |
|---|---|---|
| **Experimental models** | | |
| HK-2 | American Type Culture Collection (ATCC) | CRL-2190 |
| A-498 | American Type Culture Collection (ATCC) | HTB-44 |
| 786-O | American Type Culture Collection (ATCC) | CRL-1932 |
| NOD/SCID mice | Beijing Vital River Laboratory Animal Technology Company (Beijing, China) | 406 |
| **Antibodies** | | |
| Ki67 | Zsbio Commerce Store, China | ZA-0502 |
| p-AURKA | Cell Signaling Technology | 3079 |
| TPX2 | Cell Signaling Technology | 12245 |
| p-RB1 | Cell Signaling Technology | 8180 |
| CK19 | Cell Signaling Technology | 13902 |
| PAX8 | Cell Signaling Technology | 28556 |
| **Oligonucleotides and other sequence-based reagents** | | |
| AURKB | 5'-TCCCTGTTCGCATTCAAC CT-3' and 5'-GTCCCACTGCTATTCTCC ATCAC-3' | N/A |
| CDC45 | 5'-TTCGTGTCCGATTTCCG CAAA-3' and 5'-TGGAACCAGCGTATATT GCAC-3' | N/A |
| TPX2 | 5'-AGGGGCCCTTTGAAC TCTTA-3' and 5'-TGCTCTAAACAAGCCC CATT-3' | N/A |
| 18 S | 5'-GTAACCCGTTGAACC CCATT-3' and 5'-CCATCCAATCGGTAGT AGCG-3' | N/A |
| **Chemicals, enzymes and other reagents** | | |
| MycoSensor PCR Assay Kit (Stratagene) | Agilent Technologies (Santa Clara, CA) | 302108 |
| Cell Cycle Compound Library | TargetMol (Boston, MA) | L8100 |
| LDC000067 | MedChemExpress | HY-15878 |
| Dulbecco's Modified Eagle's medium | HyClone (Washington, D.C.) | SH30022.01 |
| Fetal bovine serum | HyClone (Washington, D.C.) | SH30071.03 |
| Penicillin/streptomycin | Gibco (Carlsbad, U.S.) | 15140122 |
| Phosphate buffered saline (PBS) | HyClone (Washington, D.C.) | SH30256.01 |
| Tumor Dissociation Kit | Miltenyi Biotec (Bergisch Gladbach, Germany) | 130-095-929 |
| Mouse Cell Depletion Kit | Miltenyi Biotec (Bergisch Gladbach, Germany) | 130-104-694 |
| CellTiter Glo reagent | Promega (Madison, U.S.) | G7570 |
| Hydrogen peroxide | Sigma-Aldrich (Burlington, U.S.) | 7722-84-1 |

| Reagent/Resource | Reference or source | Identifier or catalog number |
|---|---|---|
| EDTA buffer | Sigma-Aldrich (Burlington, U.S.) | T9285 |
| SureSelectXT Human All Exon V4 / V6 + UTR Kit | Agilent Technologies (Santa Clara, CA) | S03723424 S07604624 |
| RNeasy Mini Kit | Qiagen (Venlo, Netherlands) | 74104 |
| TruSeq Stranded RNA HT Kit | Illumina (San Diego, USA) | 15032620 |
| TranScript All-in-One First-Strand cDNA Synthesis SuperMix for RT-PCR (One-Step gDNA Removal) | TransGen Biotech, China | AT341-01 |
| PerfectStart Green qPCR SuperMix | TransGen Biotech, China | AQ601-01-V2 |
| Rabbit/mouse polymer detection system Kit | Zsbio Commerce Store, China | PV-6001, PV-6002 |
| DAB reagent Kit | Zsbio Commerce Store, China | ZLI-9017 |
| **Software** | | |
| nf-core/sarek (v3.1.2) | Garcia et al, 2020 | N/A |
| BWA-MEM (v2.2.1) | Vasimuddin et al, 2019 | N/A |
| Mutect2 (GATK v4.3.0.0) | Cibulskis et al, 2013 | N/A |
| ASCAT (v3.0.0) | Van Loo et al, 2010 | N/A |
| CNVkit (v0.9.9) | Talevich et al, 2016 | N/A |
| GISTIC2.0 (v2.0.23, with GenePattern) | Mermel et al, 2011 | N/A |
| VEP (v106) | McLaren et al, 2016 | N/A |
| SigProfilerExtractor (v1.1.21) | Islam et al, 2022 | N/A |
| NeoPredPipe (v1.1) | Schenck et al, 2019 | N/A |
| POLYSOLVER (v4) | Shukla et al, 2015 | |
| maftools (v2.18.1) | Mayakonda et al, 2018 | N/A |
| IGV (Integrative Genomics Viewer, v2.16.2) | Thorvaldsdottir et al, 2013 | N/A |
| nf-core/rnaseq (v3.10.1) | Ewels et al, 2020 | N/A |
| STAR (v2.7.10a) | Dobin et al, 2013 | N/A |
| RSEM (v1.3.1) | Li & Dewey, 2011 | N/A |
| DESeq2 (v1.42.1) | Love et al, 2014 | N/A |
| clusterProfiler (v4.8.3) | Wu et al, 2021 | N/A |
| ssGSEA (v10.1.0) | Barbie et al, 2009; Subramanian et al, 2005 | N/A |
| GOnet (v2019-07-01) | Pomaznoy et al, 2018 | N/A |
| Enrichr (Access Date: 2023-07-19) | Chen et al, 2013; Kuleshov et al, 2016; Xie et al, 2021 | N/A |
| xCell (v1.1.0) | Aran et al, 2017 | N/A |
| RSeQC (v4.0.0) | Wang et al, 2012 | N/A |
| NGSCheckMate (v1.0.0) | Lee et al, 2017 | N/A |
| **Other** | | |
| Illumina HiSeq 4000, NovaSeq 6000 or NovaSeq X | Illumina (San Diego, USA) | N/A |
| 96-well plates | Thermo Fisher Scientific (Waltham, USA) | 167425 |

## Preparation of experimental models and clinical samples

### Cell lines and reagents

Commercial cell lines were purchased from ATCC with authentication performed by the authors. Cell lines were maintained in Dulbecco's Modified Eagle's medium (DMEM, HyClone, cat #SH30022.01) supplemented with 10% fetal bovine serum (HyClone, cat #SH30071.03) and 1% penicillin/streptomycin (Gibco, cat #15140122). Mycoplasma testing was performed using the MycoSensor PCR Assay Kit (Stratagene, cat #302108). A

compound library (cat #L8100) targeting cell cycle regulators was purchased from TargetMol (Boston, MA).

### Clinical samples

Each participating institution approved the study: SingHealth Centralized Institutional Review Board (CIRB Ref: 2010/735/B and 2019/2351); Medical Ethics Committee of Tianjin Medical University, China (IRB Ref: EK2017014).

Treatment-naïve CDC patients, who underwent radical nephrectomy at the participating institutions, were included with written informed consents, if fresh-frozen tissues were available. CDC diagnosis was confirmed by 2–3 uropathologists. Histological staining, immunochemistry staining, and molecular profiling were used whenever possible to ensure accurate subtyping of the cases. All animal studies were conducted in compliance with animal protocols approved by the Institutional Animal Care and Use Committee of Sun Yat-sen University Cancer Center (Guangzhou, China). The experiments conformed to the principles set out in the WMA Declaration of Helsinki and the Department of Health and Human Services Belmont Report.

### Establishment of patient-derived xenograft (PDX) and patient-derived CDC cells

To establish the PDX model, female NOD/SCID mice (6–8 weeks old) were purchased from Beijing Vital River Laboratory Animal Technology Company (Beijing, China) and housed under specific pathogen-free conditions in the Laboratory Animal Center of Sun Yat-sen University. Tumor tissue (CDC1, distant spinal metastasis of the left renal collecting duct) was obtained from the First Affiliated Hospital of Sun Yat-sen University with the patient's informed consent, from which one PDX model and primary cell line were established. Specifically, tumor tissue was washed in PBS and minced with a sterile scalpel to patches that could pass through a needle bore. Mice were implanted subcutaneously with pieces of fresh CDC1 tumor tissue. Tumors were excised when they reached 1000 mm$^3$ to establish the patient-derived CDC cells. H&E staining was performed to confirm the morphology of the CDC tumor. CDC1 tumor cells were disassociated from the CDC1 PDX tumors by using the Tumor Dissociation Kit (Miltenyi Biotec, cat #130-095-929). To ensure the purity of patient-derived tumor cells, the mouse-derived cells were removed using the Mouse Cell Depletion Kit (Miltenyi Biotec, cat #130-104-694). The purified CDC1 cells were then seeded and maintained in DMEM/F12 (1:1) medium supplemented with 10% FBS and 1% penicillin/streptomycin. At 80 to 90% confluence, the cells were passaged at a 1:2 ratio.

## Genome and transcriptome characterization of samples

Whole-exome (WES) and transcriptome (RNA-seq) sequencing was performed on an Asian CDC cohort ($n = 14$, 13 normal-matched tumors and 1 PDX model) to identify recurrent somatic mutations, mutational signatures, and associated gene regulatory network changes. CDC RNA-seq data ($n = 9$) from Msaouel et al, (Msaouel et al, 2020b) was integrated with our transcriptome data to increase sample size and statistical power (Data ref: NCBI Sequence Read Archive PRJNA605003, 2020, normal samples; raw data from CDC tumor samples were provided by Dr. Msaouel).

### Whole exome sequencing (WES) and data analysis

For WES, samples were enriched using SureSelectXT Human All Exon V4/V6 + UTR Kit. Next-generation sequencing was performed with Illumina HiSeq 4000, NovaSeq 6000, or NovaSeq X sequencing machines, producing 100 bp or 150 bp paired-end reads of an average coverage of ~100X. Sarek (v3.1.2) (Garcia et al, 2020) was used to process WES data for variant calling. Briefly, the raw reads were trimmed and aligned to GRCh38 with BWA-MEM (v2.2.1) (Vasimuddin et al, 2019) with somatic small variants and copy number variation detected using Mutect2 (GATK v4.3.0.0) (Cibulskis et al, 2013), ASCAT (v3.0.0) (Van Loo et al, 2010)/ CNVkit (v0.9.9) (Talevich et al, 2016) and GISTIC2.0 (v2.0.23, with GenePattern) (Mermel et al, 2011) respectively. Only variants marked as PASS by Mutect2, having at least 50X coverage for the tumor sample and a minimum variant allele frequency (VAF) of 5%, were retained for further analysis. Variants were annotated using VEP (v106) (McLaren et al, 2016). Mutational signatures were extracted using SigProfilerExtractor (v1.1.21) (Islam et al, 2022). Neoantigen analysis was performed using NeoPredPipe (v1.1) (Schenck et al, 2019), with HLA types predicated by POLYSOLVER (v4) (Shukla et al, 2015). For the CDC1(PDX) sample, tumor-only variant calling was performed with a panel of normal (PoN) built from all the normal samples in our cohort. Similarly, the variants marked as PASS need to have at least 50X coverage and a minimum VAF of 5%. To further filter possible germline variants in CDC1, only non-silent variants were kept if they are not present in dbSNP. Genes involved in oncogenic pathways were analyzed using maftools (v2.18.1) (Mayakonda et al, 2018). Mapped reads were visualized using IGV (Integrative Genomics Viewer, v2.16.2) (Thorvaldsdottir et al, 2013).

### Whole transcriptome sequencing (RNA-seq) and data analysis

RNA was extracted using the RNeasy Mini Kit (Qiagen, cat #74104). TruSeq Stranded RNA HT Kit (Illumina, cat #15032620) was used to remove cytoplasmic and mitochondria rRNA and prepare RNA libraries from 1 µg of total RNA. About 60 million reads were sequenced using Illumina HiSeq 4000 or NovaSeq 6000 for each library (paired-end, 150 bp or 151 bp reads). Raw RNA-seq reads were processed with nf-core pipeline rnaseq (v3.10.1) (Ewels et al, 2020), which mapped the reads with STAR (v2.7.10a) (Dobin et al, 2013) and quantified gene expression using RSEM (v1.3.1) (Li and Dewey, 2011). Differentially expressed genes (DEGs) were identified by DESeq2 (v1.42.1) (Love et al, 2014), using adjusted $p$ value ≤0.01 and |log2 fold-change|≥ 2 as the cutoffs. Gene set enrichment analysis (GSEA) was performed using clusterProfiler (v4.8.3) (Wu et al, 2021) and ssGSEA (v10.1.0) (Barbie et al, 2009; Subramanian et al, 2005). Gene ontology of the DEGs was annotated using GOnet (v2019-07-01) (Pomaznoy et al, 2018). Enriched pathways were identified with Enrichr (Access Date: 2023-07-19) (Chen et al, 2013; Kuleshov et al, 2016; Xie et al, 2021). Cell-type enrichment analysis was performed using xCell (v1.1.0) (Aran et al, 2017). RSeQC (v4.0.0) (Wang et al, 2012) was used to assess the mapping quality of the libraries. To ensure proper pairing of the WES and RNA-seq samples, NGSCheckMate (v1.0.0) (Lee et al, 2017) was used.

### Drug candidate identification via in vitro screening

For drug screening, CDC1 cells were plated in 96-well plates and treated with 130 small molecules (1 µM) of the cell cycle regulator compound library (TargetMol, cat #L8100) for 96 h. Cell viability was measured using CellTiter Glo reagent (Promega, cat #G7570) according to the manufacturer's instructions. Inhibition rates were calculated as the cell viability of the drug-treated cells normalized to that of the DMSO-treated. Top 10 drug candidates with a high inhibition rate were further tested in one normal immortalized kidney cell line (HK-2), and two ccRCC cell lines (A-498, 786-O) to ensure low nephrotoxicity and drug specificity.

### Drug validation using in vivo models

All animal studies were conducted in compliance with animal protocols approved by the Institutional Animal Care and Use Committee of Sun Yat-sen University Cancer Center (Guangzhou, China). Five- to six-week-old NOD/SCID mice were purchased from Beijing Vital River Laboratory Animal Technology Company.

To validate drug efficacy in vivo, CDC1 (PDX) tumor masses were passaged in NOD/SCID mice after subcutaneous implantation. When the tumor volumes reached approximately 100 mm³, the mice were divided into two groups for treatment. Randomization was performed by equally dividing the tumor-bearing mice with a similar tumor burden into groups for drug treatment. The CDK9 inhibitor LDC000067 (MedChemExpress, cat # HY-15878) was suspended in 1 × saline and was given by oral gavage for 21 days (10 mg/kg daily) with blinding. Tumor volume and body weight were monitored every two days until the tumor volume reached 1000 mm³. Mice were sacrificed by $CO_2$ inhalation and the survival time of each mouse was recorded.

### Real-time qRT-PCR

Total RNA was extracted using the RNeasy Mini Kit (Qiagen, Germany, cat #74106). cDNA was subsequently produced using TranScript All-in-One First-Strand cDNA Synthesis SuperMix for RT-PCR (One-Step gDNA Removal) (TransGen Biotech, China, cat #AT341-01). qRT-PCR was conducted following the instructions of PerfectStart Green qPCR SuperMix (TransGen Biotech, China, cat #AQ601-01-V2). 18 S was used as an endogenous housekeeping gene for normalization. The primer pairs of the genes used for real-time qRT-PCR are as follows: *AURKB* 5′-TCCCTGTTCGCATTCAACCT-3′ and 5′-GTCCCAC TGCTATTCTCCATCAC-3′; *CDC45* 5′-TTCGTGTCCGATTTCCG CAAA-3′ and 5′-TGGAACCAGCGTATATTGCAC-3′; *TPX2* 5′-AG GGGCCCTTTGAACTCTTA-3′ and 5′-TGCTCTAAACAAGCCCC ATT-3′; 18 S 5′-GTAACCCGTTGAACCCCATT-3′ and 5′-CCATCCA ATCGGTAGTAGCG-3′. The mRNA levels of these genes were determined as the mean of the Ct values obtained from a couple of primers. Data were described as relative mRNA expression levels.

### Immunohistochemistry

We performed immunohistochemistry as previously described (Chen et al, 2023). Briefly, tumor tissues derived from patients or animal models were fixed with 4% formalin, embedded in paraffin, and sectioned. Paraffin sections were incubated with 3% hydrogen peroxide to block endogenous peroxidase for 15 min at 37 °C and rinsed with 0.01 M PBS, followed by high-pressure antigen retrieval in EDTA buffer. The sections were then incubated with antibodies specific to Ki67 (Zsbio Commerce Store, cat #ZA-0502), p-AURKA (Cell Signaling Technology, cat #3079), TPX2 (Cell Signaling Technology, cat #12245), p-RB1 (Cell Signaling Technology, cat

**The paper explained**

**Problem**

Collecting duct carcinoma (CDC) remains a lethal and aggressive form of kidney cancer with no effective treatment, primarily because of our limited understanding of its underlying molecular alterations and etiology. The rarity of the CDC and the lack of preclinical models, particularly for Asian patients, have hindered scientific and clinical progress in this field. This study aimed to comprehensively characterize the molecular alterations of CDC and establish preclinical models for drug screening and testing, to identify potentially targetable therapeutic vulnerabilities.

**Results**

Through genome and transcriptome sequencing, we identified frequent KRAS hotspot mutations, in addition to known TP53 and NF2 mutations. Notably, the mutational signature SBS22 caused by aristolochic acid (AA) exposure was detected in 3/13 patients. The presence of SBS22, known to be more prevalent in Asia, highlights a geologically specific etiology of the disease. We further found that cell cycle-related pathways were highly dysregulated. Using our newly established CDC preclinical models, coupled with drug screening, we identified a CDK9 inhibitor, LDC000067, as a promising candidate which could specifically target CDC tumor growth and extend survival.

**Impact**

Our study demonstrated that targeting cell-cycle machinery effectively suppressed CDC tumor growth, laying a solid foundation for initiating clinical trials aimed at treating patients affected by this highly aggressive cancer.

#8180), CK19 (Cell Signaling Technology, cat #13902) and PAX8 (Cell Signaling Technology, cat #28556) at 4 °C overnight, respectively. After rinsing 3 times in PBS, the sections were incubated with a rabbit polymer detection system kit (Zsbio Commerce Store, cat #PV-6001/PV-6002) at room temperature for 1 h. To determine the specificity of the immunostaining, IgG was used as a negative control. Then, immunoreactivity was measured using a DAB reagent kit (Zsbio Commerce Store, cat #ZLI-9017) according to the manufacturer's instructions. The immunostaining scores were assessed under a microscope by two pathologists according to the following formula: score = 3 (strongly positive) × percentage + 2 (moderately positive) × percentage + 1 (weak positive) × percentage + 0 (negative) × percentage. Percentage was defined using the following criteria: 0 (<10% positive cells), 1 (10–25% positive cells), 2 (25–50% positive cells), 3 (50–75% positive cells), and 4 (75–100% positive cells).

## For more information

- https://rarediseases.info.nih.gov/diseases/9573/collecting-duct-carcinoma
- https://www.orpha.net/en/disease/detail/247203
- https://www.malacards.org/card/collecting_duct_carcinoma

## Data availability

The dataset supporting the conclusions of this article is available in the Sequence Read Archive (SRA), under BioProject PRJNA1004347. The

datasets generated and/or analysed during the current study are available in the Research Data Deposit public platform (www.researchdata.org.cn) with approval number RDDB2024336467.

The source data of this paper are collected in the following database record: biostudies:S-SCDT-10_1038-S44321-024-00102-5.

## Peer review information

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

## Acknowledgements

We thank all patients who participated in this study. This project was supported by the National Key Research and Development Program of China (No. 2022YFA1304000) and the National Natural Science Foundation of P. R. China (82320108015, 82073391, and 82170188), Guangzhou Science and Technology Program (2022B01J1004), NCC Research Fund and SingHealth Academic Clinical Program (ACP) Grant [ref. 08/FY2018/EX(SL)/45-A86]. BTT is supported by Singapore Translational Research Investigator (STaR) Award (National Medical Research Council (NMRC), MOH-000248-00) and Tan Yew Oo Professorship in Pathology. PG was partially supported by Khoo Pre-Doctoral Fellowship (Duke-NUS Medical School) and Open Fund - Young Individual Research Grant (OF-YIRG) (NMRC, Project ID: MOH-001152-00). We thank the Laboratory Animal Center of Sun Yat-sen University for providing core facility services.

## Author contributions

**Peiyong Guan**: Conceptualization; Data curation; Formal analysis; Funding acquisition; Investigation; Visualization; Writing—original draft; Writing—review and editing. **Jianfeng Chen**: Data curation; Formal analysis; Validation; Visualization; Writing—review and editing. **Chengqiang Mo**: Data curation; Project administration. **Tomoya Fukawa**: Conceptualization; Data curation; Investigation. **Chao Zhang**: Data curation; Project administration. **Xiuyu Cai**: Data curation; Project administration. **Mei Li**: Data curation. **Jing Han Hong**: Writing—original draft; Writing—review and editing. **Jason, Yongsheng Chan**: Writing—original draft; Writing—review and editing. **Cedric Chuan Young Ng**: Data curation. **Jing Yi Lee**: Data curation. **Suet Far Wong**: Data curation. **Wei Liu**: Data curation. **Xian Zeng**: Data curation. **Peili Wang**: Project administration. **Rong Xiao**: Data curation; Project administration. **Vikneswari Rajasegaran**: Data curation. **Swe Swe Myint**: Data curation. **Abner Ming Sun Lim**: Formal analysis. **Joe Poh Sheng Yeong**: Data curation. **Puay Hoon Tan**: Data curation. **Choon Kiat Ong**: Data curation. **Tao Xu**: Data curation. **Yiqing Du**: Data curation. **Fan Bai**: Data curation. **Xin Yao**: Supervision; Project administration. **Bin Tean Teh**: Conceptualization; Supervision; Funding acquisition; Writing—review and editing. **Jing Tan**: Conceptualization; Supervision; Funding acquisition; Writing—review and editing.

Source data underlying figure panels in this paper may have individual authorship assigned. Where available, figure panel/source data authorship is listed in the following database record: biostudies:S-SCDT-10_1038-S44321-024-00102-5.

## Disclosure and competing interests statement

The authors declare no competing interest.

