## [Peer Review File · EMBO Molecular Medicine]

Comprehensive Molecular Characterization of Collecting Duct Carcinoma for Therapeutic Vulnerability

Peiyong Guan, Jianfeng Chen, Chengqiang Mo, Tomoya Fukawa, Chao Zhang, Xiuyu Cai, Mei Li, Jing Han Hong, Jason Chan, Cedric Chuan Young Ng, Jing Yi Lee, Suet Far Wong, Wei Liu, Xian Zeng, Peili Wang, Rong Xiao, Vikneswari Rajasegaran, Swe Swe Myint, Abner Ming Sun Lim, Joe PS Yeong, Puay Hoon Tan, Choon Kiat Ong, Tao Xu, Yiqing Du, Fan Bai, Xin Yao, Bin Tean Teh, and Jing Tan

Corresponding authors: Jing Tan (tanjing@sysucc.org.cn), Bin Tean Teh (teh.bin.tean@singhealth.com.sg), Xin Yao (yaoxin@tjmuch.com)

Review Timeline:

Submission Date:	19th Feb 24
Editorial Decision:	15th Apr 24
Revision Received:	16th May 24
Editorial Decision:	19th Jun 24
Revision Received:	24th Jun 24
Accepted:	1st Jul 24

Editor: Zeljko Durdevic

Transaction Report:

15th Apr 2024

Dear Prof. Tan,

Thank you for the submission of your manuscript to EMBO Molecular Medicine, and please accept my apologies for the delay in getting back to you, which is due to the fact that one referee needed more time to complete his/her review. We have now received feedback from the three reviewers who agreed to evaluate your manuscript. As you will see from the reports pasted below, all three referees recognize potential interest of the study but also raise important and partially overlapping concerns that should be addressed in a major revision of the current manuscript. If you would like to discuss further the points raised by the referees, I am available to do so via email or video. Let me know if you are interested in this option.

We would welcome the submission of a revised version within three months for further consideration. Please let us know if you require longer to complete the revision.

Please use this link to login to the manuscript system and submit your revision: <https://embomolmed.msubmit.net/cgi-bin/main.plex>

I look forward to receiving your revised manuscript.

Yours sincerely,

Zeljko Durdevic

We require:

- 1) A .docx formatted version of the manuscript text (including legends for main figures, EV figures and tables). Please make sure that the changes are highlighted to be clearly visible.
- 2) Individual production quality figure files as .eps, .tif, .jpg (one file per figure). For guidance, download the 'Figure Guide PDF': (<https://www.embopress.org/page/journal/17574684/authorguide#figureformat>).
- 3) A .docx formatted letter INCLUDING the reviewers' reports and your detailed point-by-point responses to their comments. As part of the EMBO Press transparent editorial process, the point-by-point response is part of the Review Process File (RPF), which will be published alongside your paper.
- 4) A complete author checklist, which you can download from our author guidelines (<https://www.embopress.org/page/journal/17574684/authorguide#submissionofrevisions>). Please insert information in the checklist that is also reflected in the manuscript. The completed author checklist will also be part of the RPF.
- 5) Please note that all corresponding authors are required to supply an ORCID ID for their name upon submission of a revised manuscript.

6) It is mandatory to include a 'Data Availability' section after the Materials and Methods. Before submitting your revision, primary datasets produced in this study need to be deposited in an appropriate public database, and the accession numbers and database listed under 'Data Availability'. Please remember to provide a reviewer password if the datasets are not yet public (see <https://www.embopress.org/page/journal/17574684/authorguide#dataavailability>).

13) Author contributions: You will be asked to provide CRediT (Contributor Role Taxonomy) terms in the submission system. These replace a narrative author contribution section in the manuscript.

14) A Conflict of Interest statement should be provided in the main text.

Please also suggest a striking image or visual abstract to illustrate your article as a PNG file 550 px wide x 300-800 px high.

**** Reviewer's comments ****

Referee #1 (Remarks for Author):

Guan et al present "Comprehensive Molecular Characterization of Collecting Duct Carcinoma for Therapeutic Vulnerability." The manuscript utilized several collecting duct carcinoma tumor samples to determine genomic and transcriptomic differences in normal vs tumor samples. They identified differences in cell cycle genes in tumor samples as a therapeutic vulnerability in CDC. In a PDX that was then transitioned to a cell line, the authors conducted a single dose drug screen and identified a set of inhibitors. One inhibitor was then studied in vivo where there was tolerability and efficacy seen.

Overall, there are a number of exciting aspects of this manuscript but the authors should be careful with the claim that this is the first study of Asian patients. There are a number of areas of concern in the RNAseq analyses that need further clarification.

Major:

1) Clarifications of the novel CDC1 PDX / cell line is needed. Specifically:

-The CDC1 (PDX) is a PDX model of metastasis derived from patient CDC1. This should consider separating from the analyses of other patients as it currently suggests that the sample is a primary sample and it is unclear if tumor evolution has occurred to establish the PDX/cell line and thus can be misconstrued by a reader.

-Sample labelled CDC1 (PDX) shows highest TMB, however, does not have SBS22 mutation and only 2 oncogenic mutations, LATS1 and FAT1? Can the authors discuss this?

-In Fig 3A, CDC1 (PDX) also clusters separately from other tumor samples. Can the authors show that CDC1 cells are similar to the other primary tumors a bit earlier or explain what is driving the PC2? Or the authors do this in Fig 4B and perhaps this needs to be introduced earlier

2) Line 134 and then in the discussion - looks like there is the following article published and should be incorporated into the manuscript as it represents 10 patients with normal-matched samples. There however, results seem to be different than the current manuscript. Can the authors discuss these differences?

Zhang, H., Lu, X., Huang, G. et al. A genomic mutation spectrum of collecting duct carcinoma in the Chinese population. *BMC Med Genomics* 15, 1 (2022). <https://doi.org/10.1186/s12920-021-01143-2>

3) Figure 2 / RNAseq analyses -

-Was any batch effect seen when integrating CDC data from other sources?

-Fig 2E is confusing as there are 7 samples - there are 4 RTK-RAS RNAseq sample - where is the 4th sample?

-Fig 2F should be interpreted cautiously given the limited numbers and the confidence interval should be discussed in the text at 188-189

-Then in Fig 2G-I, your Fig 2A suggests there are 3 AA samples but only 2 are shown?

-Authors claim that cell cycle is the driver of tumorigenesis. However, several of their samples (atleast 3/9 tumor samples In Fig 4B) do not show changes in G2M checkpoint pathway or E2F targets.

4) In supplementary figure 2, some samples were excluded from analysis what is the justification for not including them in future analysis. Moreover, the 3 PCA plots, look the same for each of 3 pathways analyzed. Can the authors speculate on the reason for the same or is this a typographical error? Do they have overlapping gene expression across samples and pathways.

5) In Fig 5B, the mice in the treatment group also died after dosing was stopped. Do the tumors increase in size after the treatment is removed?

6) Methods for cell line / mouse studies are quite limited and should be better explained (eg growth characteristics / media used to grow the cells, tolerability/efficacy of mouse studies and doses used etc...)

Minor

- 1) Line 158 (and throughout) - KRAS G12* may be misinterpreted for a nonsense mutation. Would spell it out like it was done in the abstract.
- 2) Line 160 - When saying the "pathway level," can the authors clarify?
- 3) The CDC1/PDX sample has a much higher TMB than the other samples, can the authors comment on this? Can the authors comment on the tumor purity of samples sequenced as supp fig 1H suggests there may be a correlation of purity and TMB?
- 4) Then for CNV changes, usually computational algorithms such as GISTIC can help to determine if there is significance to CNVs observed before making a claim there is no significant CNVs in CDCs. Have authors performed such studies? Again, the prior study from Zhang et al and others suggest some other CNVs may exist? (Lines 174-175)
- 5) Lines 193-194 - It is unclear what the authors mean and seems like this should be cited
- 6) Line 237, would be good to say "one normal immortalized kidney cell line"... as these cells were immortalized and while used for toxicology studies, are not true normal kidney cells
- 7) Supplementary figure 1A: some mutational signatures are not defined in key, eg: CDH15 in T05, FAT1, RABEP1

Referee #2 (Comments on Novelty/Model System for Author):

I think this is an interesting and well performed analysis. The medical impact may be limited due to the rarity of this tumor, however, I think this work may provide additional information on hard to treat cancers, which is where treatment options are most needed.

Referee #2 (Remarks for Author):

The paper by Guan is based on a well performed study providing insights on a rare/hard to treat tumor type; kidney collecting duct carcinoma (CDC). This cohort of patients described in the manuscript is unique as it specifically focuses on Asian patients which have not studied before. The cohort is also relatively large compared to previous analysis. The authors have used a multi-omic approach to capture the mutational and transcriptomic landscape of these tumors. In addition, the authors have also developed the first PDX model of CDC that was used to assess response to CDK9 inhibition. Given the rarity of this tumor type and its aggressive phenotype, this paper is timely and provides important information for better understanding CDC and hard to treat tumors in general. The paper is clearly written, and the results are clearly presented. I have a few questions for the authors.

1. In Fig 1 panel C, what is used as reference population? For some of the genes there seems to be an up/down regulation in both mutant and wt tumors. Are these comparisons made against normal appearing tissue? If so, the authors may want to add this information to the figure for clarity.
2. For the KRAS mutant tumors, do the authors know how these mutations translated in activation of downstream signaling molecules like Raf/MEK/ERK in the mutant compared to the WT tumors? Was this activation consistent across mutation types?
3. It is quite impressive that the authors were able to establish a new cell line and PDX model of CDC. I am curious to understand whether an attempt was made to establish these models also from other tissue samples and the degree of success. This could be an interesting information to add to understand how easily (or not) these models can be created.
4. It would be interesting to test response to CDK9i in the other 2 cell lines available and potentially correlate response to key shared genomic/transcriptomic alterations. This would further strengthen the results shown in this manuscript, although it may be difficult to have access to this material.
5. The authors seem to imply in the discussion a potential use for immunotherapies in these patients, but I am not sure the data presented necessarily point in this direction. Was this statement supported by expression levels of immune-checkpoints or any specific characterization of the immune infiltrate?

Referee #3 (Comments on Novelty/Model System for Author):

There are some concerns with technical approaches and presentation; see comments below for full details.

Referee #3 (Remarks for Author):

This is an additional CDC genomics study (10 cases) that seems relevant and should therefore be cited and discussed: PMID 34980126. The statement in the Introduction that "...CDC in Asians has not been characterized..." and a similar statement of this being the "first" study are not strictly true and should be edited.

Were the sequenced tumors from the primary tumor site or a metastatic site? This should be stated in the main text and this information could be included in Figure 1. Were all sequenced tumors treatment-naïve? Or did some tumors have chemo/IO exposure prior to sequencing? Again, this important detail should be briefly stated in the main text.

I understand that Sarek was used to identify mutations from the WES data, but what additional (if any) filtering steps were used to identify likely deleterious mutations? Or were all non-synonymous mutations included in the analysis and shown in Figure 1 and Suppl Figure 1? How were genes selected for inclusion in Figure 1 (vs Suppl Figure 1)? It is noteworthy that the most commonly mutated genes in the cohort (TTN, MUC16) are also large genes that are notorious for rising to top of mutation lists when stringent filtering is not performed.

The beginning of the Results section says 8 tumors had RNA-seq performed, but there are only 6-7 cases with RNA-seq throughout Figure 2. Please clarify.

The diagrams in Figure 2C and D are quite confusing. For example, in Figure 2C, how can "ESTROGEN_RESPONSE_LATE" be both significantly down and significantly up in Hippo WT tumors?

I understand the focus on the two tumors with AA mutational signature, but 3 tumors also had a unique signature associated with haloalkane exposure (SBS42). Why no specific analysis/discussion of those 3 tumors?

I do not agree with authors' assessment of CDC1 cell line based on PCA plot in Figure 3. Authors state that CDC1 'resembled the CDC tumors' but the PCA plot shows that CDC1 is about as dissimilar tumors as it is to normal tissues.

Supplementary Figure 2 lacks context or description. Its very hard to know what to make of this data in its current form. Why were the outliers excluded from the analysis?

The organization of data presentation is somewhat confusing to me. Figure 1 is the DNA data, Figure 2 is the RNA data in the context of the DNA data, then Figure 3 is the RNA data without context of the DNA data. It seems more natural to me to swap Figures 2 and 3 (and accompanying text). It seems weird to talk so much about the RNA-seq data in Figure 2, but be ignoring what you will say in Figure 3 is the most important aspect of the RNA data (ie, cell cycle). I kept trying to figure how the analyses in Figure 2 did NOT show cell cycle pathways...ie, why did we have to wait for the analyses in Figure 3 to see the importance of cell cycle pathway alteration?

Was the CDC1 cell line and PDX established from the same tumor site that underwent WES and RNA-seq?

Lines 244-245 should be edited from "kidney" to "a kidney cell line"

A few of the critical details regarding the PDX experiment should be included in the main text (drug dose, delivery method, duration of therapy). Were the IHC experiments performed from tumors collected at the end of the experiment? Or were a separate cohort of treated mice used for the IHC studies?

Response Letter (Manuscript # EMM-2024-19502)

Dear Reviewers

We are grateful for the chance to substantially improve our manuscript according to your constructive suggestions. In particular, to address the overlapping concerns of the referees, we performed immunohistochemistry (IHC) staining on CDC1 patient and PDX samples to confirm their collecting duct carcinoma identity.

Please allow us to take this opportunity to thank you for taking the time to review our manuscript and for your suggestions to improve it. We believe that our revised manuscript has adequately addressed all issues raised and is ready for re-review.

Yours Sincerely,

Jing Tan, PhD
Bin Tean Teh, M.D., PhD

P.S.

Summary of Revisions:

1. Immunohistochemistry staining of CDC1 patient and PDX tissues to confirm their collecting duct carcinoma identity.
2. GISTIC2.0 analysis for the focal copy number changes.
3. Comparing our results with "*Zhang H et al (2022a), A genomic mutation spectrum of collecting duct carcinoma in the Chinese population. BMC Med Genomics 15: 1*".
4. Other specific changes suggested by the reviewers.
5. Format changes required by the journal.

Color keys:

- **Black** text: reviewers' comments.
- **Blue** text: authors' responses.
- **Red** text: changed made to text in manuscript.

***** Reviewer's comments *****

Referee #1 (Remarks for Author):

Guan et al present "Comprehensive Molecular Characterization of Collecting Duct Carcinoma for Therapeutic Vulnerability." The manuscript utilized several collecting duct carcinoma tumor samples to determine genomic and transcriptomic differences in normal vs tumor samples. They identified differences in cell cycle genes in tumor samples as a therapeutic vulnerability in CDC. In a PDX that was then transitioned to a cell line, the authors conducted a single dose drug screen and identified a set of inhibitors. One inhibitor was then studied in vivo where there was tolerability and efficacy seen.

Overall, there are a number of exciting aspects of this manuscript but the authors should be careful with the claim that this is the first study of Asian patients. There are a number of areas of concern in the RNAseq analyses that need further clarification.

Response: Thank you for the comments and are happy you shared our excitement.

Major:

1) Clarifications of the novel CDC1 PDX / cell line is needed. Specifically:

-The CDC1 (PDX) is a PDX model of metastasis derived from patient CDC1. This should consider separating from the analyses of other patients as it currently suggests that the sample is a primary sample and it is unclear if tumor evolution has occurred to establish the PDX/cell line and thus can be misconstrued by a reader.

Response: Thank you for the suggestion and we have removed CDC1 (PDX) from **Fig 1** and **Fig 3**. Only tumor samples were included in these two figures. CDC1 will be introduced instead in **Fig 4** for drug screening, which is more logical. To confirm identity of the CDC1 (PDX), we have detected the CDC markers, such as PAX8 and CK19 in CDC1 patient and PDX samples using immunohistochemistry (IHC) staining. The results showed that strong positive expression of PAX8 and CK19 in both samples, providing compelling evidence supporting their authenticity as CDC (**Response Fig 1**). The **Response Fig 1** is also provided in **Fig 4B**.

We have amended the main text by adding the follow: "To confirm the identity of the PDX (CDC1) sample as CDC, we first conducted immunohistochemistry (IHC) staining for PAX8 and CK19 in both CDC1 patient and PDX samples. The results showed that strong positive

expression of PAX8 and CK19 in both samples, indicating these samples are real CDC (**Fig 4B**)”.

Additionally, we performed ssGSEA analysis and confirmed CDC1 transcriptome showed similar profiles with CDC tumors and is different from normal kidney.

For completeness, variants called for CDC1 (PDX) are now provided in **Appendix Table S3**.

-Sample labelled CDC1 (PDX) shows highest TMB, however, does not have SBS22 mutation and only 2 oncogenic mutations, LATS1 and FAT1? Can the authors discuss this?

Response: Thank you for pointing out this issue. The relatively higher TMB of CDC1 (PDX) was because, unlike the other 13 normal-matched tumors, it has no matched normal tissue or blood to filter away the germline variants. Variants of CDC1 (PDX) were differently called using all normal samples in our cohort as a panel of normal (PoN), therefore, potentially have variants that are present in the patient’s germline. As a result, these variants are not 100% somatic, one of the reasons why CDC1 has no SBS22, despite of the high mutation

counts. To avoid confusions, we have excluded CDC1 (PDX) from **Fig 1**. Variants for CDC1 (PDX) are now provided in **Appendix Table S3** for completeness.

-In Fig 3A, CDC1 (PDX) also clusters separately from other tumor samples. Can the authors show that CDC1 cells are similar to the other primary tumors a bit earlier or explain what is driving the PC2? Or the authors do this in Fig 4B and perhaps this needs to be introduced earlier

Response: Thank you for pointing out the issue. It is expected that the CDC1 (PDX) cluster away from other tumors in PC2, because the CDC1 cells used for RNA-seq were isolated from the CDC1 PDX tumor and were more pure cancer cells than the other 13 primary tumor samples, which contain cells other than cancer cells. Hence, it is no longer appropriate to show them in the same PCA plot for the purpose of showing their similarity. To avoid further confusion, we excluded CDC1 (PDX) from **Fig 3A**, since it was primarily used for drug screening and testing, which is logically falls under **Fig 4** and **5**.

2) Line 134 and then in the discussion - looks like there is the following article published and should be incorporated into the manuscript as it represents 10 patients with normal-matched samples. There however, results seem to be different than the current manuscript. Can the authors discuss these differences?

Zhang, H., Lu, X., Huang, G. et al. A genomic mutation spectrum of collecting duct carcinoma in the Chinese population. *BMC Med Genomics* 15, 1 (2022). <https://doi.org/10.1186/s12920-021-01143-2>

Response: Thank you for pointing us to this paper, which came out after we did the initial literature review. We should have searched the literature again prior to submission. We have cited the paper and contrasted their findings with ours.

Following suggestions from another reviewer, we performed GISTIC2.0 analysis on our copy number data. We identified one significantly deleted region at 9p21.3, which agree with findings in Zhang *et al*. It is worth pointing out that this deleted region contains cell-cycle related genes CDKN2A and CDKN2B (q-value = 0.071) (**Appendix Figure S1J, K**). But the expression changes of these two gene were not reduced but increased significantly in CDC tumors (**Appendix Figure S1L, M**). Hence the role of the focal deletion in CDC tumorigenesis remains to be investigated.

We also included the following in the first paragraph of the “Discussion” section.

“To date, only one other genomic study focused on Chinese CDC patients (n = 10) using formalin-fixed, paraffin-embedded (FFPE) samples, which found similar TMB with ours (1.37 vs our 1.86 per MB) (Zhang et al, 2022a). Zhang et al. identified many focal amplifications and deletions, among which 9p21.3, where CDKN2A and CDKN2B genes are located, is the only focal deletion found in our cohort. However, given the discordant gene expression changes in CDKN2A and CDKN2B, role of 9p21.3 deletion in CDC tumorigenesis remains to be investigated. Such extensive focal changes in Zhang et al. cohort might be related to the nature of the FFPE samples. In terms of somatic small variants, TP53 was the only common top frequent mutations, but with different frequency (10% versus 23% in our cohort). The low concordance of the genomic changes indicates the heterogeneity of CDC tumorigenesis, which requires orthogonal characterizations, such as transcriptomic profiling.”

3) Figure 2 / RNAseq analyses -

-Was any batch effect seen when integrating CDC data from other sources?

Response: Thank you for pointing out this issue. Because of the technical differences between the two cohorts, we did not merge the two cohorts of RNAseq data, instead we separately analyzed them and confirmed the concordance between the fold-changes of the genes. Thus, batch-effect is not an issue.

-Fig 2E is confusing as there are 7 samples - there are 4 RTK-RAS RNAseq sample - where is the 4th sample?

Response: Thank you for pointing out this issue. **Appendix Figure S2B**, the sample T10 was excluded from the analysis, as it clustered together with the RTK-RAS wt samples. We are selective in the samples to mitigate the small sample size. Similarly, we also excluded a few other samples from the analysis, based on the overall transcriptome profiles. The main text and figure legends have been modified to make this sample exclusion clearer.

-Fig 2F should be interpreted cautiously given the limited numbers and the confidence interval should be discussed in the text at 188-189

Response: Thank you for pointing out this issue. We highlighted the sample size limitation in the second paragraph of the “Discussion” section, by adding the following.

“Because of our small sample size, and the wide confident interval especially for KRAS wildtype patients, how or whether the KRAS hotspot mutations lead to poorer survival requires further investigations.”

-Then in Fig 2G-I, your Fig 2A suggests there are 3 AA samples but only 2 are shown?

Response: Thank you for pointing out this issue. Like the RT-KRAS case above, we excluded outlier samples from the analysis based on the overall transcriptome profiles of the tumors, to mitigate the sample size issue (circled in **Appendix Figure S2**). We have modified the text and figure legends to be explicit about the sample selection.

-Authors claim that cell cycle is the driver of tumorigenesis. However, several of their samples (atleast 3/9 tumor samples In Fig 4B) do not show changes in G2M checkpoint pathway or E2F targets.

Response: Thank you for raising this issue. We agree with your observation and hypothesize this primarily reflects the tumor heterogeneity. These tumors, such as T10 may represent a different subset of CDC tumors. With more accumulated samples and evidence, CDC like other kidney cancers could be further divided into distinct subgroups.

We have modified the main text to the following to highlight the heterogeneity.

“We next confirmed that CDC1 cell line transcriptionally resembled CDC tumors, distinct from normal kidneys, though some of the tumors, such as T10, showed relatively lower level of cell-cycle pathways, highlighting tumor heterogeneity (**Fig 4C**).”

4) In supplementary figure 2, some samples were excluded from analysis what is the justification for not including them in future analysis. Moreover, the 3 PCA plots, look the same for each of 3 pathways analyzed. Can the authors speculate on the reason for the same or is this a typographical error? Do they have overlapping gene expression across samples and pathways.

Response: Thank you for pointing out these issues. The PCA plots are similar, because they are based on the transcriptome of the same set of samples. The difference is that samples in different panels were highlighted by their mutation status in Hippo and RTK-RAS pathways or presence/absence of AA mutational signature. The purpose of these PCA plots is to show how the different mutations overlap with the transcriptome profile of the tumors.

We used these plots mainly to exclude outlier samples to mitigate the heterogeneity and small sample size. We have modified the figure legend to the following to explain clearer.

“Circled samples were excluded from comparisons, because they did not cluster transcriptomically with majority of the samples sharing the same mutated pathway. This allowed us to study relatively more homogeneous groups of samples.”

5) In Fig 5B, the mice in the treatment group also died after dosing was stopped. Do the tumors increase in size after the treatment is removed?

Response: Thank you for pointing out this issue. We observed increase in tumor size after the treatment is removed. When the size of tumors reached 1,000 mm³, the survival dates were recorded.

6) Methods for cell line / mouse studies are quite limited and should be better explained (eg growth characteristics / media used to grow the cells, tolerability/efficacy of mouse studies and doses used etc...)

Response: Thank you for the constructive suggestion. We have modified the “Materials and methods” section to include the following required details on cell line and PDX experiments.

“To validate drug efficacy in vivo, CDC1 (PDX) tumor masses were passaged in NOD/SCID mice after subcutaneous implantation. When the tumor volumes reached approximately 100 mm³, the mice were divided into two groups for treatment. Randomization was performed by equally dividing the tumor-bearing mice with a similar tumor burden into groups for drug treatment. The CDK9 inhibitor LDC000067 was suspended in 1 × saline and was given by oral gavage for 21 days (10mg/kg daily) without blinding. Tumor volume and body weight were monitored every two days until the tumor volume reached 1,000 mm³. Mice were sacrificed by CO₂ inhalation and the survival time of each mouse was recorded.”

The full details are also available in the Sections 1 and 2 of the **Appendix Materials and Methods 1**.

Minor

1) Line 158 (and throughout) - KRAS G12* may be misinterpreted for a nonsense mutation. Would spell it out like it was done in the abstract.

Response: Thank you for pointing out this issue. We have amended the main text accordingly.

2) Line 160 - When saying the "pathway level," can the authors clarify?

Response: Thank you for pointing out this issue. We were referring to the percentage of patients with mutations in the genes involved in the Hippo signalling pathway. To be clearer, we have revised the main text to “When somatic mutations are grouped by the oncogenic pathways they are involved in”.

3) The CDC1/PDX sample has a much higher TMB than the other samples, can the authors comment on this? Can the authors comment on the tumor purity of samples sequenced as supp fig 1H suggests there may be a correlation of purity and TMB?

Response: Thank you for pointing out this issue. The relatively higher TMB of CDC1 was because it has no matched tumor tissue or blood to filter away all germline variants. Instead, the variants were called using all normal samples as a panel of normal (PoN), thus, potentially have variants that are present in the patient's germline.

Because of the difference in variant calling methods, mutations in CDC1/PDX are no longer comparable to the other 13 normal-matched CDC tumors. To avoid confusions, we have excluded CDC1 (PDX) from **Fig 1** and provided the mutations in **Appendix Table S3**.

Tumor purity indeed impacts small variants and copy number calling. We checked and found no statistically significant correlation between the tumor purity and TMB ($p = 0.71$, **Response Fig 2**)

purity and tumor mutational burden (TMB).

4) Then for CNV changes, usually computational algorithms such as GISTIC can help to determine if there is significance to CNVs observed before making a claim there is no significant CNVs in CDCs. Have authors performed such studies? Again, the prior study from Zhang et al and others suggest some other CNVs may exist? (Lines 174-175)

Response: Thank you for the suggestion and we have performed GISTIC2.0 analysis on our copy number data. We identified one significantly deleted region at 9p21.3, which agree with the findings from Zhang *et al*. It is worth pointing out that this deleted region contains cell-cycle related genes CDKN2A and CDKN2B (q-value = 0.071) (**Response Fig 3A, B**). But the expression changes of these two gene were not reduced but increased significantly in CDC tumors (**Response Fig 3C, D and Appendix Figure S1L, M**). Hence the role of the focal deletion in CDC tumorigenesis remains to be investigated.

We have included the following into the results section.

“No focal amplification was identified whereas focal deletion was found at 9p21.3 (q-value = 0.071), where CDKN2A and CDKN2B genes are located, though with marginal statistical significance. (**Appendix Figure S1I, J and K**). However, focal deletion in the 9p21.3 did not reduce expression levels of CDKN2A or CDKN2B, as they were increased by 3.37 and 1.39 log2-folds in the tumors respectively (**Appendix Figure S1L and M**).”

5) Lines 193-194 - It is unclear what the authors mean and seems like this should be cited

Response: Thank you for pointing out this issue. We have cited the following paper on AA mutational signatures.

Poon, S. L., Pang, S. T., McPherson, J. R., Yu, W., Huang, K. K., Guan, P., Weng, W. H., Siew, E. Y., Liu, Y., Heng, H. L., Chong, S. C., Gan, A., Tay, S. T., Lim, W. K., Cutcutache, I., Huang, D., Ler, L. D., Nairismagi, M. L., Lee, M. H., . . . Teh, B. T. (2013). Genome-wide mutational signatures of aristolochic acid and its application as a screening tool. *Sci Transl Med*, 5(197), 197ra101. <https://doi.org/10.1126/scitranslmed.3006086>

6) Line 237, would be good to say "one normal immortalized kidney cell line"... as these cells were immortalized and while used for toxicology studies, are not true normal kidney cells

Response: Thank you for the suggestion. We have amended it accordingly.

7) Supplementary figure 1A: some mutational signatures are not defined in key, eg: CDH15 in T05, FAT1, RABEP1

Response: Thank you for pointing out this issue. We have corrected the keys in the legends.

Referee #2 (Comments on Novelty/Model System for Author):

I think this is an interesting and well perform analysis. The medical impact may be limited due to the rarity of this tumor, however, I think this work may provide additional information on hard to treat cancers, which is were treatment options are most needed.

Referee #2 (Remarks for Author):

The paper by Guan is based on a well performed study providing insights on a rare/hard to treat tumor type; kidney colleting duct carcinoma (CDC). This cohort of patients described in the manuscript is unique as it specifically focuses on Asian patients which have not studied before. The cohort is also relatively large compared to previous analysis. The authors have used a multi-omic approach to capture the mutational and transcriptomic landscape of these tumors. In addition, the authors have also developed the first PDX model of CDC that was used to assess response to CDK9 inhibition. Given the rarity of this tumor type and its aggressive phenotype, this paper is timely and provides important information for better understanding CDC and hard to treat tumors in general. The paper is clearly written, and the results are clearly presented. I have a few questions for the authors.

Response: Thank you for your complement and positive feedback.

1. In Fig 1 panel C, what is used as reference population? For some of the genes there seems to be an up/down regulation in both mutant and wt tumors. Are these comparisons made against normal appearing tissue? If so, the authors may want to add this information to the figure for clarity.

Response: Thank you for pointing out this issue. We believe you were referring to **Fig 2C**. The reference population is the 4,384 genes in the Hallmark gene set collection, the default setting of clusterProfiler, which was used for enrichment analysis.

For gene sets that are both up- and down- regulated when comparing the same conditions (e.g., ESTROGEN_RESPONSE_LATE in **Fig 2C** when comparing Hippo wt tumor with normal), the up- and down-regulated genes are non-overlapping set of different genes in the pathway, indicating genes in the pathway are not changing expressions in the same direction.

We have amended the legend of **Fig 2** to make it clearer.

2. For the KRAS mutant tumors, do the authors know how these mutations translated in activation of downstream signaling molecules like Raf/MEK/ERK in the mutant compared to the WT tumors? Was this activation consistent across mutation types?

Response: Thank you for pointing out this issue. KRAS mutations, frequently observed in colorectal cancer, lung cancer, multiple myeloma and pancreatic cancer, were found to be allele- and tissue-specific (Cook et al., Nat Commun 12, 1808 (2021)). Downstream the KRAS hotspot mutation (G12A/D/V) in kidney cancer, such as CDC, is less known and requires further investigation. **Fig 2E** presented 16 up-regulated (KRAS mut vs wt tumors) known KRAS_SIGNALING genes, based on our small sample size, which are also up-regulated when comparing with normal samples. These genes are likely the downstream changes upon KRAS mutation. Among these genes, INHBA and PTGS2 are also part of the HALLMARK_TNFA_SIGNALING_VIA_NFKB pathway. Thus, the signalling upon KRAS mutation is unclear with our limited sample size.

3. It is quite impressive that the authors were able to establish a new cell line and PDX model of CDC. I am curious to understand whether an attempt was made to establish these models also from other tissue samples and the degree of success. This could be an interesting information to add to understand how easily (or not) these models can be created.

Response: Thank you for the compliment. We did not establish CDC cell line and PDX models from other tissue samples, which are difficult to access due to the rarity of this disease. However, we plan to establish these models from other CDC tissue samples in our further studies, through our close collaborations with the clinicians.

4. It would be interesting to test response to CDK9i in the other 2 cell lines available and potentially correlate response to key shared genomic/transcriptomic alterations. This would further strengthen the results shown in this manuscript, although it may be difficult to have access to this material.

Response: Thank you for your constructive suggestion. We had earlier planned to validate our CDK9i in the other two CDC cell lines. However, we had issue accessing them. Although we tried to get in touch with the author for the cell lines, we have not received a reply. We will try to establish collaboration with them when the opportunity comes. CDC tumors will be routinely collected by our clinician collaborators. And in the lab, we will try to establish new cell lines and PDXs, which can serve as the resources for future drug screening and testing.

5. The authors seem to imply in the discussion a potential use for immunotherapies in these patients, but I am not sure the data presented necessarily point in this direction. Was this statement supported by expression levels of immune-checkpoints or any specific characterization of the immune infiltrate?

Response: Thank you for pointing out this issue. Immunotherapies are mostly applicable to the AA+ tumors in our patient cohort. There are two folds of reasons: 1) AA+ tumors tend to have higher neo-antigen presentation (**Fig 1F**) and 2) there are relatively higher CD8+ naïve T-cells and Th1 cells in the AA+ tumors in our cohort (**Fig 2I**).

Referee #3 (Comments on Novelty/Model System for Author):

There are some concerns with technical approaches and presentation; see comments below for full details.

Referee #3 (Remarks for Author):

This is an additional CDC genomics study (10 cases) that seems relevant and should therefore be cited and discussed: PMID 34980126. The statement in the Introduction that "...CDC in Asians has not been characterized..." and a similar statement of this being the "first" study are not strictly true and should be edited.

Response: Thank you for pointing us to this paper. Sorry that we missed it because it was published after we drafted our manuscript. We should have conducted another round of literature review. Our cohort differs from this paper (Zhang *et al.*) in a few aspects, but not limited to: 1) our cohort have matched transcriptome data, allowing us to investigate the downstream implications of the somatic mutations; and 2) Zhang *et al.* cohort was from FFPE samples. Ours are fresh-frozen tissue, which makes the variant calling less challenging and more accurate.

We have cited Zhang *et al.* and added the following comparisons in the "Discussion" section of the revised manuscript.

"To date, only one other genomic study focused on Chinese CDC patients (n = 10) using formalin-fixed, paraffin-embedded (FFPE) samples, which found similar TMB with ours (1.37 vs our 1.86 per MB) (Zhang *et al.*, 2022a). Zhang *et al.* identified many focal amplifications and deletions, among which 9p21.3, where CDKN2A and CDKN2B genes are located, is the only focal deletion found in our cohort. However, given the discordant gene expression changes in CDKN2A and CDKN2B, role of 9p21.3 deletion in CDC tumorigenesis remains to be investigated. Such extensive focal changes in Zhang *et al.* cohort might be related to the nature of the FFPE samples. In terms of somatic small variants, TP53 was the only common top frequent mutations, but with different frequency (10% versus 23% in our cohort). The low concordance of the genomic changes indicates the heterogeneity of CDC tumorigenesis, which requires orthogonal characterizations, such as transcriptomic profiling."

Were the sequenced tumors from the primary tumor site or a metastatic site? This should be stated in the main text and this information could be included in Figure 1. Were all sequenced tumors treatment-naïve? Or did some tumors have chemo/IO exposure prior to sequencing? Again, this important detail should be briefly stated in the main text.

Response: Thank you for pointing out these issues. Tumors T01 to T13 are all treatment-naïve primary tumors. There was no chemo/IO exposure prior to sequencing. CDC1 PDX and cell line were from distant (spinal) metastasis from the left renal collecting duct.

We have included the following detailed diagnosis information of both primary and metastatic tumors of CDC1 in **Appendix Table S1**.

- Primary tumor:

Immunohistochemistry: CK(+), Vim(+), CK(H, partially +), CK(L+), CAM5.2(+), EMA(+), CK19(+), CK18(+), CK8 (+), CD15 (few cells +), , Ki67 (about 2%+), CD34 (blood vessels +), D2-40 (lymphatic vessels +), CD10(-), Villin(-), CK7(-).

Diagnosis: (Left kidney) Microscopic histomorphological changes combined with immunohistochemistry results are consistent with renal Bellini collecting duct carcinoma, and the cancer tissue invades the perirenal adipose tissue.

- Metastatic tumor:

Immunohistochemistry: Heterocyst, CK(+), PAX8(+), CK(HMW)(-), TTF-1(-), CDX-2(-), CK7 (-).

Diagnosis: Based on the HE morphology, immunohistochemistry and medical history, the lesion is consistent with metastasis of renal cell carcinoma (thoracic spine). Please refer to the pathology report of the primary tumor for tumor classification.”

We have also modified the “Materials and method” section to reflect the characteristics of the samples.

I understand that Sarek was used to identify mutations from the WES data, but what additional (if any) filtering steps were used to identify likely deleterious mutations? Or were all non-synonymous mutations included in the analysis and shown in Figure 1 and Suppl Figure 1? How were genes selected for inclusion in Figure 1 (vs Suppl Figure 1)? It is noteworthy that the most commonly mutated genes in the cohort (TTN, MUC16) are also large genes that are notorious for rising to top of mutation lists when stringent filtering is not performed.

Response: Thank you for pointing out this issue. For the 13 normal-matched fresh-frozen tissues, besides only keeping variants within the “SureSelect Human All Exon V6+UTR” (S07604624) and marked as PASS by Mutect2, we did not further filter the somatic

mutations called. However, to make sure the variants called are real and not resulted from spurious mappings to repeat regions, we performed manual verification of all variants by checking the reads mapping in IGV (Integrative Genomics Viewer).

We have added the following to the “Appendix Materials and Methods”.

“Only variants within the “SureSelect Human All Exon V6+UTR” kit (S07604624) and marked as PASS by Mutect2 were retained for further analysis.”

The CDC1(PDX) has no matched normal/blood samples. Therefore, tumor-only variant calling was performed with panel of normal (PoN) built from all the normal samples in our cohort. Further filtering was performed to exclude germline variants as much as possible.

We have added the following to the “Appendix Materials and Methods”.

“To further filter possible germline variants in CDC1, we require the tumor sample to have at least 50X coverage and a minimum variant allele frequency (VAF) of 5%. Only non-silent variants were kept if they are not present in dbSNP.”.

The TTN and mucin genes were shown for completeness purpose. We have modified the figure and legend to highlight this. These genes are also greyed out to avoid over-interpretation.

The beginning of the Results section says 8 tumors had RNA-seq performed, but there are only 6-7 cases with RNA-seq throughout Figure 2. Please clarify.

Response: Thank you for highlighting this and sorry for the confusions caused. We have clarified by adding the following into the main text.

“To mitigate our relatively small sample size, we excluded outlier samples that did not cluster transcriptomically with majority of the samples sharing the same mutated pathway. This allowed us to study relatively more homogeneous groups of samples (**Appendix Figure S2**).”

The diagrams in Figure 2C and D are quite confusing. For example, in Figure 2C, how can "ESTROGEN_RESPONSE_LATE" be both significantly down and significantly up in Hippo WT tumors?

Response: In **Fig 2C**, the 38 down-genes and 36 up-genes in Hippo WT tumors are non-overlapping. It is similar for **Fig 2D (Response Table 1)**. Such observation indicates

estrogen response genes are not changing in the same direction upon the mutations of interest. Of note, many cell cycle genes (highlighted in bold) are up regulated.

Response Table 1. ESTROGEN_RESPONSE_LATE genes in Fig 2C and D

Hippo Wt (Fig 2C)		RTK-RAS Mut (Fig 2D)	
Down (n = 38)	Up (n = 36)	Down (n = 31)	Up (n = 30)
ACOX2	AREG	ACOX2	AGR2
ALDH3A2	CAV1	ANXA9	AREG
ANXA9	CCNA1	ASS1	CAV1
ASS1	CCND1	CA12	CD44
CA12	CD44	CA2	CDC20
CA2	CDC20	CACNA2D2	CDC6
CACNA2D2	CDC6	CALCR	CLIC3
CALCR	CLIC3	CISH	FABP5
CISH	DLG5	CKB	GINS2
CKB	FABP5	CXCL14	GJB3
CXCL12	GINS2	DNAJC12	ISG20
CXCL14	GJB3	FGFR3	KIF20A
DNAJC12	ISG20	FRK	KRT19
FGFR3	KIF20A	HSPB8	LAMC2
FRK	KRT19	IL17RB	LSR
HMGCS2	LAMC2	IMPA2	MDK
HSPA4L	LSR	KCNK5	MYOF
HSPB8	MDK	MAPT	PERP
IGSF1	MICB	NPY1R	PKP3
IL17RB	MYOF	PCP4	PLK4
IMPA2	PERP	PDZK1	PTGES
KCNK5	PKP3	PGR	RAPGEFL1
MAPT	PLK4	PRLR	S100A9
NPY1R	PTGES	PTGER3	SFN
PCP4	RNASEH2A	RET	SLC16A1
PDZK1	S100A9	SEMA3B	SLC7A5
PGR	SFN	SERPINA5	STIL
PRLR	SLC16A1	SLC22A5	TFAP2C
PTGER3	SLC7A5	SLC27A2	TOP2A
RET	STIL	SLC9A3R1	TSPAN13
SCUBE2	TFAP2C	TNNC1	
SEMA3B	TFPI2		
SERPINA5	TMPRSS3		
SLC22A5	TOP2A		
SLC27A2	TPBG		
SLC9A3R1	TRIM29		
TJP3			
TNNC1			

I understand the focus on the two tumors with AA mutational signature, but 3 tumors also had a unique signature associated with haloalkane exposure (SBS42). Why no specific analysis/discussion of those 3 tumors?

Response: Thank you for your constructive suggestion. For T02 and T11 has lower percentages of SBS42 (32% and 29% respectively) than their SBS1 (aging, clock-like). Though T08 has relatively higher percentage (81%) of SBS42 but has a relatively lower TMB, compared to AA samples. These variations are unlike AA+ tumors, where AA mutational signature is the most dominate signature across all three samples (>65%).

Considering these variations in the percentages of SBS42, as we do not have occupational data on the patients, the role of haloalkane exposure on CDC tumorigenesis still requires further study, although SBS42 was found in printing worker who developed cholangiocarcinoma (Alexandrov et al, 2020, Nature 578: 94-101). It is well recognized that occupational exposures to carcinogens, such as haloalkane, is important for cancer prevention.

Thus, we have amended the “Discussion” section as below to highlight the importance and limitations of our study.

“As we do not have occupational data on patients, the role of haloalkane exposure on CDC tumorigenesis requires further study, although it was previously found in printing worker who developed cholangiocarcinoma (Alexandrov et al, 2020).”

I do not agree with authors' assessment of CDC1 cell line based on PCA plot in Figure 3. Authors state that CDC1 'resembled the CDC tumors' but the PCA plot shows that CDC1 is about as dissimilar tumors as it is to normal tissues.

Response: Thank you for pointing this out. In hindsight, we agree with you and feel that it is inappropriate to cluster tumor samples (consisting multiple cell types) with the cell line (mostly cancer cells), making the presentation of the results misleading. Additionally, CDC1 has no matched normal tissue or blood to filter away the germline variants. Instead, its variants were called using all normal samples in our cohort as a panel of normal (PoN), therefore, potentially have variants that are present in the patient's germline. To avoid confusions, we have excluded CDC1 from PCA plot in **Fig 3** and **Fig 1**. CDC1 is now characterized in **Fig 4** when we do the drug screening. For completeness, the variants called are now provided in **Appendix Table S3**.

To confirm identity of the CDC1 (PDX), we have detected the CDC markers, such as PAX8 and CK19 in CDC1 patient and PDX samples by immunohistochemistry (IHC) staining. The results showed that strong positive expression of PAX8 and CK19 in both samples, providing compelling evidence supporting their authenticity as CDC (**Response Fig 1**). The **Response Fig 1** is also provided in **Fig 4B**.

We have amended the main text by adding the follow: “To confirm the identity of the PDX (CDC1) sample as CDC, we first conducted immunohistochemistry (IHC) staining for PAX8 and CK19 in both CDC1 patient and PDX samples. The results showed that strong positive expression of PAX8 and CK19 in both samples, indicating these samples are real CDC (**Fig 4B**)”.

Supplementary Figure 2 lacks context or description. Its very hard to know what to make of this data in its current form. Why were the outliers excluded from the analysis?

Response: Thank you for pointing out this issue. We have amended legend of **Appendix Figure S2** and included the following in the main text.

“To mitigate our relatively small sample size, we excluded outlier samples that did not cluster transcriptomically with majority of the samples sharing the same mutated pathway. This allowed us to study relatively more homogeneous groups of samples (**Appendix Figure S2**).”

The organization of data presentation is somewhat confusing to me. Figure 1 is the DNA data, Figure 2 is the RNA data in the context of the DNA data, then Figure 3 is the RNA data without context of the DNA data. It seems more natural to me to swap Figures 2 and 3 (and accompanying text). It seems weird to talk so much about the RNA-seq data in Figure 2, but be ignoring what you will say in Figure 3 is the most important aspect of the RNA data (ie, cell cycle). I kept trying to figure how the analyses in Figure 2 did NOT show cell cycle pathways...ie, why did we have to wait for the analyses in Figure 3 to see the importance of cell cycle pathway alteration?

Response: Thank you for your comments and suggestions.

We previously considered different orderings of the figures. Since **Fig 3** is about identification of the cell cycle pathway, **Fig 4** (drug screening) and **Fig 5** (validations) logically follows.

Fig 2 is about transcriptome changes in the context of somatic mutations. It serves two purposes: 1) panels B, C, D and H in **Fig 2** implied the involvement of cell cycle pathway, as shown by the enrichment of G2M_CHECKPOINT, E2F_TARGETS and MITOTIC_SPINDLE pathways; and 2) it also demonstrated the heterogeneity of the CDC tumors, implying the necessity to identify a potentially targetable commonly dysregulated downstream pathway, which is what **Fig 3** is about.

We do agree with you that the current text did not flow very well from **Fig 2** to **Fig 3**. Therefore, we have amended the main text by adding more linkers like below to smooth the transition.

“Therefore, we hypothesized that, although CDC tumors exhibit heterogeneity in terms of somatic mutations and associated transcriptomic dysregulation, cell cycle-related pathways may be a commonly induced pathway in CDC that is potentially targetable. We next studied cell-cycle pathways in CDC in depth.”

Was the CDC1 cell line and PDX established from the same tumor site that underwent WES and RNA-seq?

Response: Yes, the CDC1 cell line and PDX were established from the same tumor site that underwent WES and RNA-seq.

Lines 244-245 should be edited from "kidney" to "a kidney cell line"

Response: Thank you for pointing out this issue. We have amended it.

A few of the critical details regarding the PDX experiment should be included in the main text (drug dose, delivery method, duration of therapy). Were the IHC experiments performed from tumors collected at the end of the experiment? Or were a separate cohort of treated mice used for the IHC studies?

Response: Thank you for your suggestion. We have added the following details of the PDX experiments in the “Materials and methods” section.

“To validate drug efficacy in vivo, CDC1 (PDX) tumor masses were passaged in NOD/SCID mice after subcutaneous implantation. When the tumor volumes reached approximately 100 mm³, the mice were divided into two groups for treatment. Randomization was performed by equally dividing the tumor-bearing mice with a similar tumor burden into groups for drug

treatment. The CDK9 inhibitor LDC000067 was suspended in 1 x saline and was given by oral gavage for 21 days (10mg/kg daily) without blinding. Tumor volume and body weight were monitored every two days until the tumor volume reached 1,000 mm³. Mice were sacrificed by CO₂ inhalation and the survival time of each mouse was recorded.”

A separate cohort of treated mice were used for the IHC studies.

19th Jun 2024

Dear Prof. Tan,

Thank you for the submission of your revised manuscript to EMBO Molecular Medicine. I am pleased to inform you that we will be able to accept your manuscript pending the following final amendments:

- 1) Address all the referee #1 concerns. No additional experiments are required. Acceptance of the manuscript will depend on the completeness of your responses included in the next, final version of the manuscript. For this reason, and to save you from any frustrations in the end, I would strongly advise against returning an incomplete revision.
- 2) In the main manuscript file, please do the following:
 - Please address all comments suggested by our data editors listed below:
 - o Figure legends:
 1. Please note that the exact p values are not provided in the legends of figures 2i; 3c-d.
 2. Please indicate the statistical test used for data analysis in the legends of figures 2c-d, h; 3d.
 3. Although 'n' is provided, please describe the nature of entity for 'n' in the legends of figures 4d; 5a, c, e-h.
 4. Please note that the error bars are not defined in the legend of figure 5c.
 - The manuscript sections should be in the following order: Title page - Abstract & Keywords - Introduction - Results - Discussion - Methods - Data Availability - Acknowledgments - Disclosure Statement & Competing Interests - References - Figure Legends - Tables with legends - Expanded View Figure Legends.
 - Rename "Conflict of interest" to "Disclosure and competing interests statement". We updated our journal's competing interests policy in January 2022 and request authors to consider both actual and perceived competing interests. Please review the policy <https://www.embopress.org/competing-interests> and update your competing interests if necessary. Also, remove the sentence "The authors declare no conflict of interests." from page 4.
 - Author contributions: Please remove it from the manuscript and specify author contributions in our submission system. CRediT has replaced the traditional author contributions section because it offers a systematic machine-readable author contributions format that allows for more effective research assessment. You are encouraged to use the free text boxes beneath each contributing author's name to add specific details on the author's contribution. More information is available in our guide to authors: <https://www.embopress.org/page/journal/17574684/authorguide#authorshipguidelines>
 - All Materials and Methods need to be described in the main text. We would encourage you to use 'Structured Methods', our new Methods format. According to this format, the Methods section should include a Reagents and Tools Table (listing key reagents, experimental models, software and relevant equipment and including their sources and relevant identifiers) followed by a Methods and Protocols section in which we encourage the authors to describe their methods using a step-by-step protocol format with bullet points, to facilitate the adoption of the methodologies across labs. More information on how to adhere to this format as well as downloadable templates (.docx) for the Reagents and Tools Table can be found in our author guidelines: <https://www.embopress.org/page/journal/17574684/authorguide#structuredmethods>
 - An example of a Method paper with Structured Methods can be found here: <https://www.embopress.org/doi/full/10.1038/s44320-024-00037-6#sec-4>
 - Indicate in legends number and nature of replicates and exact p= values, not a range, along with the statistical test used. To keep the figures "clear" some authors found providing an Appendix table Sx with all exact p-values preferable. You are welcome to do this if you want to.
 - In Methods, provide the statement that the informed consent was obtained from all human subjects and that the experiments conformed to the principles set out in the WMA Declaration of Helsinki and the Department of Health and Human Services Belmont Report.
- 3) Appendix: Please combine all Appendix figures and tables S1 and S2 into a single PDF file and add table of content with page numbers on the title page. Appendix methods should be moved to the main manuscript file. Appendix table S3 should be renamed to Dataset EV1 and uploaded as such. Please updated its callouts in the main manuscript text. Remove Appendix figure legends from the main manuscript file and place them under their corresponding figures.
- 4) Funding: Information about funding should be placed in "Acknowledgments", thus please remove "Financial support" from page 3. Please make sure that information about all sources of funding are complete in both our submission system and in the manuscript. Currently, Khoo Pre-Doctoral Fellowship and MOH-OFYIRG21 nov-0002 are missing in our submission system.
- 5) The Paper Explained: Please add it to the main manuscript text.
- 6) Synopsis: Every published paper now includes a 'Synopsis' to further enhance discoverability. Synopses are displayed on the journal webpage and are freely accessible to all readers. They include separate synopsis image and synopsis text.
 - Synopsis image: Please provide a striking image or visual abstract as a high-resolution jpeg file 550 px-wide x (250-400)-px high to illustrate your article.
 - Synopsis text: Please provide a short standfirst (maximum of 300 characters, including space) as well as 2-5 one sentence bullet points that summarise the paper as a .doc file. Please write the bullet points to summarise the key NEW findings. They should be designed to be complementary to the abstract - i.e. not repeat the same text. We encourage inclusion of key acronyms and quantitative information (maximum of 30 words / bullet point). Please use the passive voice.
 - Please check your synopsis text and image before submission with your revised manuscript. Please be aware that in the proof

stage minor corrections only are allowed (e.g., typos).

7) For more information: This space should be used to list relevant web links for further consultation by our readers. Could you identify some relevant ones and provide such information as well? Some examples are patient associations, relevant databases, OMIM/proteins/genes links, author's websites, etc...

8) As part of the EMBO Publications transparent editorial process initiative (see our Editorial at <http://embomolmed.embopress.org/content/2/9/329>), EMBO Molecular Medicine will publish online a Review Process File (RPF) to accompany accepted manuscripts. This file will be published in conjunction with your paper and will include the anonymous referee reports, your point-by-point response and all pertinent correspondence relating to the manuscript. Let us know whether you agree with the publication of the RPF and as here, if you want to remove or not any figures from it prior to publication. Please note that the Authors checklist will be published at the end of the RPF.

9) Please provide a point-by-point letter INCLUDING my comments as well as the reviewer's reports and your detailed responses (as Word file).

I look forward to reading a new revised version of your manuscript as soon as possible.

Yours sincerely,

Zeljko Durdevic

*** Instructions to submit your revised manuscript ***

1) a .docx formatted version of the manuscript text (including Figure legends and tables)

2) Separate figure files*

3) supplemental information as Expanded View and/or Appendix. Please carefully check the authors guidelines for formatting Expanded view and Appendix figures and tables at <https://www.embopress.org/page/journal/17574684/authorguide#expandedview>

4) a letter INCLUDING the reviewer's reports and your detailed responses to their comments (as Word file).

5) The paper explained: EMBO Molecular Medicine articles are accompanied by a summary of the articles to emphasize the major findings in the paper and their medical implications for the non-specialist reader. Please provide a draft summary of your article highlighting

6) For more information: There is space at the end of each article to list relevant web links for further consultation by our readers. Could you identify some relevant ones and provide such information as well? Some examples are patient associations, relevant databases, OMIM/proteins/genes links, author's websites, etc...

7) Author contributions: the contribution of every author must be detailed in a separate section.

8) EMBO Molecular Medicine now requires a complete author checklist (<https://www.embopress.org/page/journal/17574684/authorguide>) to be submitted with all revised manuscripts. Please use the checklist as guideline for the sort of information we need WITHIN the manuscript. The checklist should only be filled with page numbers where the information can be found. This is particularly important for animal reporting, antibody dilutions (missing) and exact values and n that should be indicated instead of a range.

9) Every published paper now includes a 'Synopsis' to further enhance discoverability. Synopses are displayed on the journal webpage and are freely accessible to all readers. They include a short stand first (maximum of 300 characters, including space) as well as 2-5 one sentence bullet points that summarise the paper. Please write the bullet points to summarise the key NEW findings. They should be designed to be complementary to the abstract - i.e. not repeat the same text. We encourage inclusion of key acronyms and quantitative information (maximum of 30 words / bullet point). Please use the passive voice. Please attach these in a separate file or send them by email, we will incorporate them accordingly.

You are also welcome to suggest a striking image or visual abstract to illustrate your article. If you do please provide a jpeg file 550 px-wide x 300-800px high.

10) A Conflict of Interest statement should be provided in the main text

11) Please note that we now mandate that all corresponding authors list an ORCID digital identifier. This takes <90 seconds to complete. We encourage all authors to supply an ORCID identifier, which will be linked to their name for unambiguous name identification.

Currently, our records indicate that the ORCID for your account is 0000-0002-4605-4624.

Link Not Available

Photos 400-800 DPI

*Additional important information regarding figures and illustrations can be found at

<https://bit.ly/EMBOPressFigurePreparationGuideline>. See also figure legend preparation guidelines:

<https://www.embopress.org/page/journal/17574684/authorguide#figureformat>

***** Reviewer's comments *****

Referee #1 (Comments on Novelty/Model System for Author):

I am most concerned about the filtering steps for some of the subanalyses in Fig S2A-C. Novelty remains high with the functional work. The model system used needs a bit more characterization (this was there before but I think during the revision process, inadvertently removed).

Referee #1 (Remarks for Author):

Many thanks to the authors for their thoughtful responses. The authors have addressed many points raised. While improved, there are several areas to consider for clarity:

Major

1, The filtering step of what samples to keep in downstream analyses remains a bit confusing. In Fig S2A, the PC1 was 31.7 and PC2 was 18.8. The authors seem to focus on comparisons based on PC1 so then they filter out samples that don't "fit" the PCA. Perhaps not filtering and basing the analyses solely on the mutational status will decrease the number of significant hits but be more reflective of the data presented?

2, If 9p21.3 is deleted and includes CDKN2A/B (I assume the authors confirmed these genes were indeed deleted), how is expression significantly increased in the CDC tumors? It may be important to clarify which samples had focal deletion (looking at IGV or equivalent). It seems like T10 and T13 are driving the increased expressions of CDKN2A/B so likely have these genes retained?

3, With shifting figures and text around, I am concerned that the genomic characterization of the CDC1 PDX/primary cell line has been lost (grossly, the transcriptional data is shown in 4C). I can't find any information in the Supp Figures. It is important to help convince the readers that these new models are faithful with known genomics that the authors present - at a minimum, being able to show the genomics of the patient/pdx/primary cells would be helpful. This would allow future researchers determine if the model generated may be helpful for their studies (if a Hippo vs Ras vs TP53 mutated). Apologies if I missed it and it is somewhere else?

Minor

1, sotorasib should be lowercase as it is the compound name and not brand name (line 319)

Referee #3 (Remarks for Author):

Is suitable for publication

Referee #1 (Remarks for Author):

Many thanks to the authors for their thoughtful responses. The authors have addressed many points raised. While improved, there are several areas to consider for clarity:

Response: we greatly appreciate you taking the time to review our manuscript again and provide your insightful comments and suggestions below.

Major

1, The filtering step of what samples to keep in downstream analyses remains a bit confusing. In Fig S2A, the PC1 was 31.7 and PC2 was 18.8. The authors seem to focus on comparisons based on PC1 so then they filter out samples that don't "fit" the PCA. Perhaps not filtering and basing the analyses solely on the mutational status will decrease the number of significant hits but be more reflective of the data presented?

Response: Thank you for your comments and suggestion.

We agree with the reviewer that an unsupervised approach without pre-filtering of outlier samples would be ideal for uncovering potential novel findings, as these excluded outliers may represent different cancer subtypes.

In fact, we had analyzed the data with and without outlier samples. It was found that inclusion of outlier samples significantly disrupted the identification of differentially expressed genes, resulting in reduced enrichment of dysregulated pathways. Specifically,

- When retaining the 3 outlier samples in Fig S2A, there was no more enrichment in the differentially expressed genes identified.
- For Fig S2B, although we could still identify KRAS_SIGNALING_UP (adjusted p-value = 0.004) and EPITHELIAL_MESENCHYMAL_TRANSITION (adjusted p-value = 0.015), since there was only one outlier sample (T10), their p-values were much less significant than those when T10 was excluded (adjusted p-value = 2.16E-06 and 6.97E-06 respectively). Additionally, including the outlier sample missed the TNFA_SIGNALING_VIA_NFKB pathway.
- If we included the 2 outlier samples in Fig S2C, when comparing AA+ vs. AA- tumors, the only enrichment detected was COAGULATION (adjust p-value = 0.03), rather than the relevant TNFA_SIGNALING_VIA_NFKB pathway identified when the outliers were excluded.

In summary, the main reason for filtering these outliers were because of patient heterogeneity, in this case, transcriptome heterogeneity under similar genomic changes, as the outliers made it more challenging to identify differentially expressed genes, given our small sample size.

Lastly, when we accumulate a reasonably larger CDC cohort, this data could be re-analyzed to understand more about their heterogeneity for precision treatment.

2, If 9p21.3 is deleted and includes CDKN2A/B (I assume the authors confirmed these genes were indeed deleted), how is expression significantly increased in the CDC tumors? It may be important to clarify which samples had focal deletion (looking at IGV or equivalent). It

seems like T10 and T13 are driving the increased expressions of CDKN2A/B so likely have these genes retained?

Response: Thank you for your comments and questions raised. To address them, we examined copy number predictions from another tool (CNVkit) and visualized read coverage in IGV as orthogonal validations. It is also worth highlighting that we used whole exome sequencing for detecting the copy number changes, which carries its own limitation in sensitivity.

The wide peak boundaries predicated by GISTIC2.0 was chr9:21,552,810-26,844,440, which, as highlighted in the main text, had marginally significant q-value of 0.071. Using CNVkit (v0.9.9), we visualized the somatic copy number at this region, which was found to be shallow loss in most patients except for T07, T11 and T12, supporting the predicated focal loss by GISTIC2.0 (**Response Figure 1A**). We also examined the read coverage using IGV and found tumor samples had relatively lower coverage, as compared to the normal controls, further supporting the shallow deletion (**Response Figure 1B**).

Therefore, the focal deletion at 9p21.3 is not a complete loss, explaining why CDKN2A and CDKN2B are still expressed. As for why they showed elevated gene expression, especially in T10 and T13, even in the presence of shallow deletions, we hypothesize the increased gene expressions were downstream dysregulation caused by mutations in oncogenic pathways (e.g., Hippo, RTK-RAS, NOTCH, TP53, etc), consistent with the induced cell-cycle pathway observed.

We have included **Response Figure 1B** as **Appendix Figure S1L** and amended the main text by adding the following:

- In the Results section: “Read coverage at the CDKN2A and CDKN2B loci showed that tumor samples had relatively lower coverage, as compared to the normal samples, supporting shallow deletions (Appendix Figure S1L).”; and
- In the Discussion section: “We hypothesize the increased CDKN2A and CDKN2B gene expressions were downstream dysregulation caused by mutations in oncogenic pathways, consistent with the induced cell-cycle pathway observed.”

3, With shifting figures and text around, I am concerned that the genomic characterization of the CDC1 PDX/primary cell line has been lost (grossly, the transcriptional data is shown in 4C). I can't find any information in the Supp Figures. It is important to help convince the readers that these new models are faithful with known genomics that the authors present - at a minimum, being able to show the genomics of the patient/pdx/primary cells would be helpful. This would allow future researchers determine if the model generated may be helpful for their studies (if a Hippo vs Ras vs TP53 mutated). Apologies if I missed it and it is somewhere else?

Response: Thank you for raising these concerns, which we completely agree.

CDC1 carries mutations in the FAT1 and LATS1 genes of the Hippo pathway. While we excluded CDC1 from Figure 1A, we have included the complete list of mutations in the Dataset EV1 (originally **Appendix Table S3**). To further convince the readers that CDC1 model is faithful, we added the following the main text to highlight its mutations in the Hippo pathway: “CDC1 carries missense mutations in the FAT1 and LATS1 genes of the Hippo pathway (**Dataset EV1**).”

Minor

1, sotorasib should be lowercase as it is the compound name and not brand name (line 319)

Response: Thank you for pointing this out. We have corrected it accordingly.

1st Jul 2024

Dear Prof. Tan,

We are pleased to inform you that your manuscript is accepted for publication and is now being sent to our publisher to be included in the next available issue of EMBO Molecular Medicine.

Yours sincerely,
